# Polarization of beliefs as a consequence of the COVID-19 pandemic: The case of Spain

**Javier Bernacer**[1]*, **Javier García-Manglano**[2], **Eduardo Camina**[1,3], **Francisco Güell**[1]

**1** Mind-Brain Group, Institute for Culture and Society (ICS), University of Navarra, Pamplona, Spain, **2** Youth in Transition Group, Institute for Culture and Society (ICS), University of Navarra, Pamplona, Spain, **3** Faculty of Education and Psychology, University of Navarra, Pamplona, Spain

* jbernacer@unav.es

## Abstract

Spain was, together with Italy, the first European country severely affected by the COVID-19 pandemic. After one month of strict lockdown and eight weeks of partial restrictions, Spanish residents are expected to have revised some of their beliefs. We conducted a survey one year before the pandemic, at its outbreak and during de-escalation (N = 1706). Despite the lockdown, most respondents tolerated being controlled by authorities, and acknowledged the importance of group necessities over individual rights. However, de-escalation resulted in a belief change towards the intrusiveness of authorities and the pre-eminence of individual rights. Besides, transcendental beliefs–God answering prayers and the existence of an afterlife–declined after the outbreak, but were strengthened in the de-escalation. Results were strongly influenced by political ideology: the proportion of left-sided voters who saw authorities as intrusive greatly decreased, and transcendental beliefs prevailed among right-sided voters. Our results point to a polarization of beliefs based on political ideology as a consequence of the pandemic.

## Introduction

On January 7, 2020, a new coronavirus was identified in China. On January 30, in view of the first cases diagnosed outside China, the World Health Organization (WHO) declared a Public Health Emergency of International Concern. On February 11, the new virus was named SARS-CoV-2, as was the disease it causes, COVID-19. On March 11, the WHO declared a pandemic. In Spain, national authorities had been tracking suspicious cases of pneumonia since late January, with the first official case, a German tourist in La Gomera (Canary Islands) diagnosed on February 1. By February 26, the first locally transmitted infection was declared in Seville. The number of confirmed cases [1] climbed from 84 on March 1 to 6,391 on March 14, when the State of Emergency, with compulsory confinement, was declared. Daily cases peaked on March 25, with 9,630 new diagnoses. For deaths, the highest daily toll was recorded on April 2, when 961 people died. By early July, more than 250,000 infections and 28,368 deaths had been officially confirmed–mortality monitoring (MOMO) indicators point at an excess mortality of over 45,000 between March and June 2020. Strict restrictions on movement stayed

**Competing interests:** The authors have declared that no competing interests exist.

in place in all of Spain between March 14 and early May. Between May 2 and June 22, a de-escalation program was put in place, determining the conditions for a progressive easing of restrictions (by age, by schedule, by region).

At the time data were collected in this research project (March 16 –April 2, 2020, and May 3 –June 2, 2020), Spain was one of the most affected countries in the world, and the trend of the pandemic was discouraging. For example, cases were nearly multiplied by 10 between March 17[th], 2020 and April 2[nd], 2020 (from 12,868 to 124,328). During May, the curve was much steeper, but there were 21,532 new cases (from 224,501 to 246,033), despite the total lockdown (https://covid19.who.int/table). Concerning death toll, there were 354 confirmed casualties by March 16[th], 10,725 by April 2[nd], and 29,100 by June 2[nd]. These figures were only similar to those of Italy and China. Thus, Spain was one of the first and most affected countries by the COVID-19 pandemic.

In this article, we show period effects on beliefs during different stages of the COVID-19 pandemic in Spain. Past research shows that extraordinary circumstances such as crises, emergencies or natural disasters can shake the ground on which people's beliefs are rooted. These exogenous shocks can lead to the development of both prosocial (which strengthen ties to the community) or antisocial beliefs (which create divisions and suspicions between people and groups), which hence influence attitudes and behaviors [2]. On the negative side, unexpected events involving a threat beyond the individual's control give rise to epistemic, existential and social imbalances. These imbalances are ordinarily addressed through psychological adjustments aimed at restoring *meaning*–understanding the context and the environment–, *safety*–recovering some sense of control over one's circumstances–and a sense of *belonging*–a positive image of oneself and the social group [3]. On the positive side, extraordinary circumstances that affect large swathes of the population might elicit feelings of community and higher levels of within-group loyalty [4], altruism and general trust [5], and trust in political and safety institutions [6]. Research based on previous health crises such as the 2009 H1N1 (swine flu) pandemic confirm that shared threats can improve attitudes towards the government and medical organizations, which in turn increase adherence to health recommendations and guidelines [7, 8]. In this case, US citizens highly rated the quality of communication by officials, especially President Obama, and this significantly increased trust in government actions and the need for vaccination [7]. In Switzerland, a longitudinal study showed that trust in medical organizations was associated with perceived efficacy of protection measures, as well as a positive attitude towards vaccination [8]. Interestingly, a follow-up study by the same authors reported a change in attitudes towards more negative views: trust in official institutions decreased over time, and respondents were increasingly afraid of the negative or unknown effects of the vaccine. More generally, a review on attitudinal determinants of protective behaviors during the H1N1 pandemic revealed that perceived susceptibility and perceived severity of the disease, state anxiety and greater trust in authorities mainly predicted protective behaviors [9]. In conclusion, even though H1NI crisis was far less severe than COVID-19 pandemic, research points to longitudinal changes on beliefs due to the global crisis, and the importance of such attitudes in protective behaviors, including voluntary vaccination.

Attitudes and beliefs can change over time. In ordinary circumstances, adjustments follow from persuasive messages, role playing or the acquisition of new information, among other influences [10, 11]. Extraordinary and traumatic events can also trigger change in one's global beliefs and attitudes [12]. According to Jeffrey C. Alexander, this can be considered a 'cultural trauma': due to a horrible event, a collectivity may change its present and future identity, which is essential for social responsibility and political action [13]. Similarly, Kai Erikson's view on modern disasters shows how social beliefs may change in extraordinary (negative) circumstances, like the mass conversion to Christianity of Ojibwa Indian reservation (in Canada)

after the 'Spanish influenza' outbreak [14]. Whereas changes in religious beliefs may be a consequence of different types of trauma [15, 16], they can also influence how people cope with life stressors, as proposed by Pargament's theory of religious coping [17]. A pandemic has the potential to produce widespread changes in people's belief systems, since its effects are felt both at the individual level (since a majority of the population experiences the contagion or death of a close friend or relative) and at the aggregate level (since the entire population is affected greatly through legislation to stay at home, a lockdown of the economy, restrictions to movement and interactions in public places, the general feeling of threat and the information on the high prevalence of infections and the extraordinary death toll). Importantly, the impact of the pandemic on groups, cohesion and conflict (i.e. prosocial behaviors, intergroup division, social conflict) has been spotted as one research priority domain for psychological science during the COVID-19 crisis [18].

Our research question is whether the attitude of Spanish residents towards social, spiritual and interpersonal affairs has been affected by the worst global crisis in the last decades. Our hypothesis, following previous research on H1N1 pandemic and social trauma, is that confidence in authorities, transcendental and prosocial beliefs will be strengthened. However, given the actual polarization of Spanish society in terms of political ideology [19], we predict that belief changes will be strongly influenced by individual political preference.

## Materials and methods

### Samples

The procedure was revised and approved by the Committee of Ethics in Research of the University of Navarra (protocol number 2018.191). No personal information was collected, and therefore informed consent was not necessary. This project started in 2018; hence, its initial purpose was obviously unrelated with COVID-19. The goal was to analyze the belief system at a group level with network theory. To do so, we designed a 90-item survey (in Spanish) and connections between beliefs were analyzed with co-occurrence matrices. Given the sudden outbreak of the COVID-19 pandemic, and the unprecedented lockdown of the Spanish population, we selected 12 transcendental, social and interpersonal propositions from the 90-item initial survey. They were chosen to capture attitudes that could either change or remain stable during the pandemic, depending on the participant's worldview (for example, "God answers people's prayers" was expected to remain stable in a deeply religious person, whereas opinions about "Government authorities are intrusive" were expected to change due to the lockdown). This item selection was done to reduce completion time and encourage participation. In any case, participants were recruited by the same means in each time point: by diffusion in social networks and instant messaging services. Data were collected in three different time points or waves: 1) in February 2019, long before the announcement of the COVID-19 pandemic; 2) from March 16 to April 2, 2020, right after the declaration of the lockdown in Spain (March 14); 3) from May 3 to June 2, 2020, right after the commencement of the 'de-escalation', that is, the progressive relaxation of the lockdown.

As explained below, assessment in the first time point was different to the procedure in the remaining waves. Before the COVID-19 pandemic, we disseminated the 90-item survey with the goal to collect 120 responses, which was considered appropriate for an expected network of 60 nodes. In total, 156 participants (89 female, 57%) answered the electronic survey. Answers were recorded even though volunteers left the survey before completion. For that reason, the number of participants varied among items in this first wave. More precisely, item 7 was answered by 156 participants, items 1, 2 and 8 by 144, items 3 and 5 by 138, items 9 and 12 by 134, item 11 by 123, items 4 and 6 by 117, and item 10 by 114 volunteers. It should be taken

into account that, at this point, we did not expect to repeat the assessment at different time points or waves.

With the outbreak of the COVID-19 pandemic in Spain, we decided to assess whether beliefs might change during this extremely unique situation. Expecting an intense but short crisis, an evaluation in two further waves were designed: right after the outbreak, and during de-escalation (i.e. progressive relaxation of the lockdown). Assessments in waves 2 and 3 were identical: 12 items were selected from the initial list of 90, and data were recorded only if the whole survey was completed. Initially, 1182 complete surveys were collected in the second wave. After data depuration (elimination of repeated cases and of responses from outside of Spain), 1109 responses (699 from female participants, 63%) were included in this time point. With respect to the de-escalation, 475 complete answers were collected, and 441 (233 female, 52.8%) were finally included in the study after data depuration. Other sociodemographic data were collected from volunteers, including whether they personally knew someone diagnosed with COVID-19 (waves 2 and 3), and whether a close relative had passed away due to COVID-19 (wave 3) (Table 1). In this case, sample size was guided by time: we collected as many responses as we could in a time window that could be informative for our research. Since we wanted to evaluate the effect of the pandemic outbreak on beliefs, we restricted the second wave to 15 days after the declaration of the state of alarm (i.e. total lockdown). With respect to de-escalation, strict measures were progressively relaxed every two weeks; thus, we collected responses during 4 weeks (two phases in the de-escalation). Therefore, sample size was not estimated after previous reports nor effect sizes, but upon the research questions we wanted to answer.

**Table 1. Sociodemographic data of the samples included in the study.**

|  | Before COVID-19 | Outbreak | De-escalation | Spanish population[*] |
|---|---|---|---|---|
| **N** | 156 | 1109 | 441 | |
| **Female** | 89 (57%) | 699 (63%) | 233 (52.8%) | 51% |
| **Age** | | | | |
| **18–30** | 74 (47.4%) | 500 (45.1%) | 108 (24.5%) | 16.4% |
| **31–40** | 39 (25%) | 186 (16.8%) | 93 (21.1%) | 16% |
| **41–50** | 23 (14.7%) | 182 (16.4%) | 139 (31.5%) | 18.1% |
| **51–60** | 16 (10.3%) | 140 (12.6%) | 57 (12.9%) | 17.8% |
| **60+** | 4 (2.6%) | 101 (9.1%) | 44 (10%) | 29.8% |
| **Civil status** | | | | |
| **Single** | 66 (58.9%) | 590 (53.5%) | 169 (38.3%) | |
| **Married** | 43 (38.4%) | 412 (37.4%) | 231 (52.4%) | |
| **Domestic partner** | 2 (1.8%) | 57 (5.1%) | 20 (4.5%) | |
| **Divorced** | 1 (0.9%) | 31 (2.8%) | 14 (3.2%) | |
| **Widowed** | 0 | 13 (1.2%) | 7 (1.6%) | |
| **Political preference** | | | | |
| **Right parties** | 65 (59.1%) | 545 (49.1%) | 261 (59.2%) | 36.2% |
| **Left parties** | 25 (22.7%) | 396 (35.7%) | 118 (26.8%) | 41.2% |
| **Other** | 20 (18.2%) | 168 (15.2%) | 62 (14%) | 22.5% |
| **Sick acquaintance** | - | 500 (45.2%) | 145 (33%) | |
| **Deceased relative** | - | - | 37 (8.4%) | |

[*]Data on sex and age distribution is taken from the Spanish National Institute of Statistics, updated July 1[st], 2020. Data on political preference is taken from latest national elections (November 10, 2019). PP and VOX are considered right parties, PSOE and Unidas-Podemos are considered left parties, and Ciudadanos (a liberal centrist party), together with nationalist parties, are considered 'other'. Reference data on civil status is not provided because this variable is not discussed throughout the manuscript.

Considering the reference data on Spanish population provided in Table 1, the samples included in our research were representative in terms of gender distribution, although female participants were overrepresented in wave 2. With respect to age, in general terms, older participants are underrepresented in all waves. Finally, according to the latest elections, right-sided and left-sided voters are overrepresented and underrepresented in our study, respectively. Statistical analyses were carried out on 'raw' data as described below. Besides, in order to correct the unbalance in sex, age and political preference of our sample with respect to the nation totals, analyses were also replicated in a weighted database (see Supplementary Methods in S1 Text for a detailed description of iterative proportional fitting).

In waves 2 and 3, participants were also asked to generate an anonymous code that they could remember, including the final letter of their social security number, their mother's last name's initial, and the first letter of the city wherein they were born. This code, together with demographic information, was used to link the responses between these two time points, and to analyze them longitudinally. Ninety-seven participants (sex: 52 female; age: 33 18–31 yr, 18 31–40 yr, 27 41–50 yr, 10 51–60 yr, 9 60+ yr; civil status: 42 single; 51 married; 1 widowed; 1 divorced; 2 domestic partner; political preference: 60 right-side voters; 25 left-side voters; 12 'other') were identified.

All datasets (main or 'raw' data, weighted data and longitudinal data) are uploaded in Stata and csv format (compressed as ZIP: S1 Datasets).

## Survey

A set of questions was presented electronically, using Google Forms. The survey in waves 2 and 3 included 12 items or propositions as follows (originally in Spanish): 1) "I think that any failure can lead to a catastrophe"; 2) "I think that there is nothing beyond death"; 3) "I think that the world is about to end"; 4) "I think that government authorities tend to be intrusive and controlling"; 5) "I think that scientific progress can help us overcome death and live forever"; 6) "I think that individual rights are more important than the needs of any group"; 7) "I think that all human beings deserve respect"; 8) "I think that God answers people's prayers"; 9) "I think that one should help those who are weak and cannot help themselves"; 10) "I think that being controlled or dominated by others is intolerable"; 11) "I think that most people generally have good intentions"; 12) "I think that it is okay to use animals for medical research". Participants were given 5 options to express their level of agreement or disagreement with the proposition, based on our theoretical operationalization of belief [20]: 1) "I agree, and I would continue to agree even if I were shown 'irrefutable' proof to the contrary"; 2) "I agree, although I could change my mind if I were shown strong evidence"; 3) "I neither agree nor disagree"; 4) "I disagree, although I could change my mind if I were shown strong evidence"; 5) "I disagree, and I would continue to disagree even if I were shown 'irrefutable' proof". Therefore, answering 1 or 5 entails a strong commitment with or against the proposition, respectively, whereas responding 2 or 4 points to an initial agreement or disagreement with the item, respectively, although open to re-evaluation. According to our theoretical framework [20], a belief is: (1) a proposition that is taken to be true; and (2) which the subject is willing to hold even if irrefutable evidence were hypothetically argued against it. In the current study, believing in a proposition is equivalent to expressing a strong agreement with it (answering 1), and a belief in the negation of the proposition is the same as a strong disagreement with it (answering 5). Finally, having an opinion for or against a proposition is equivalent to expressing agreement (answering 2) or disagreement (answering 4) with it.

In addition, the following sociodemographic data were asked: sex (male or female), age range (18–30, 31–40, 41–50, 51–60, 61 or older), and civil status (single, married, domestic

partner, divorced, widowed). We also asked the ideology of the political parties they usually voted for: 'right-sided', 'left-sided', 'both', and 'center' options were offered, but participants could also freely type their response. In waves 2 and 3, they were asked the following: "Do you personally know someone who has been diagnosed with COVID-19?". Besides, in wave 3, we included the following question: "Has any of your close relatives passed away due to COVID-19?". Note that, in Spain, not all suspicious COVID-19 cases were officially tested with PCR or serological analyses when data were collected. Since the main focus of the current manuscript is psychological (that is, the impact of the pandemic on personal beliefs), our only interest with respect to these two questions was the subjective interpretation of each participant, and not whether diagnoses of ill acquaintances or deceased relatives were officially confirmed. In other words, if the participant considered that someone they knew had the disease, or that a close relative died of COVID-19, even in the absence of an official diagnosis, we took it as a positive response.

## Statistical analyses

Responses to each item of the survey ranged from 1 to 5, as a proxy of disagreement level (1 = strong agreement, that is, absence of disagreement; 2 = agreement, that is, weak disagreement; 3 = neutrality; 4 = disagreement; 5 = strong disagreement). For that reason, we took responses as ordinal variables, and higher values pointed to stronger disagreement. Thus, we conducted ordinal logistic analyses to test whether the selected independent variables could significantly predict responses to each item of the survey. All statistical analyses were performed in Stata IC 16 (StataCorp, College Station, TX, released in 2019).

The primary questions of our research were: 1) Are responses affected by the pandemic (outbreak and de-escalation)? 2) Is this affectation modulated by political ideology?; 3) Do responses change in participants with a COVID-19 sick acquaintance? These questions were answered with two sets of 12 ordinal logistic regressions (one for each item of the survey), which included 'response' as dependent variable (i.e. disagreement level). The first set of regressions aimed to assess the main effect of 'wave' on 'response' (question 1 above), and included the following independent variables of interest: main effects of wave (0 = before COVID-19; 1 = outbreak; 2 = de-escalation), politics (0 = right-sided voter; 1 = left-sided voter; 2 = other), and reporting a COVID-19 sick acquaintance (0 = no, 1 = yes). The second set of regressions was intended to analyze the effect of political ideology on the impact of the pandemic on 'response'; thus, the interaction between wave and politics was included. In both sets of regressions, the following covariates were included: sex (0 = male; 1 = female), age group (0 = 18–30; 1 = 31–40; 2 = 41–50; 3 = 51–60; 4 = 60+), civil status (0 = single; 1 = divorced; 2 = widowed; 3 = domestic partner; 4 = married), and reporting a COVID-19 deceased relative (0 = no, 1 = yes). Throughout the manuscript, in the main text and supplementary tables, we report odds ratios, standard errors, 95% confidence intervals, z statistics and probabilities. Since coefficients from interactions are difficult to interpret, we used the "margins" command in Stata to calculate differences in predicted probabilities [21]. Let us consider, for instance, the interaction between wave and political preference: based on the ordinal logistic regression models, "margins" computes the probability predicted by the model of a participant with certain political preference to show certain level of disagreement (1 to 5) in certain wave with respect to another time point. The effect of covariates (sex, age group and civil status) on responses was also estimated from these ordinal logistic regressions, and is summarized in S1 Text and S2 Table.

The longitudinal dataset (97 unequivocally identified participants in the outbreak and de-escalation) was analyzed with ordinal logistic mixed models (one for each item), which allow

the attribution of a between-subject variance to fixed factors, and a within-subject variance to random effects (several measures for each individual). In our case, the fixed-factor equation included the same predictors as in the cross-sectional dataset, although some variables were recoded due to the relatively small sample size. Hence, politics was transformed into a 'right-voter' variable (= 0, left-sided or ambiguous voter; = 1, right-sided voter), age was recoded as 'older' (= 0, 18–40 yr; = 1, over 41), and civil status was transformed into 'single' (= 0, married/domestic partner; = 1, single). The random-effects equation included wave (= 0, outbreak; = 1, de-escalation) nested within subject (1 to 97 for each respondent).

Statistical reports (log files from Stata) for all analyses are uploaded as (S1 File).

## Results

The main goal of this research is to assess whether the endorsement of several propositions changed as a consequence of the COVID-19 pandemic, and the influence of political preference on this possible change. In order to put these results into context, we report in S1 Table the proportion of participants that strongly agreed (1), agreed (2), were neutral (3), disagreed (4) or strongly disagreed (5) with each item, irrespective to the time point when information was collected (i.e. wave). See also S1 Text for a general description of the data in terms of political preference and sociodemographic groups.

### Effect of the pandemic on personal beliefs

Our main interest was to assess whether the different stages of the COVID-19 epidemic ('outbreak' and 'de-escalation') had an impact on the level of agreement towards the propositions presented to participants (Fig 1). Additionally, in the next section, we explore whether this possible impact was different depending on the political preference of respondents. To answer the first question, we employed an ordinal logistic regression for each item, including 'wave'

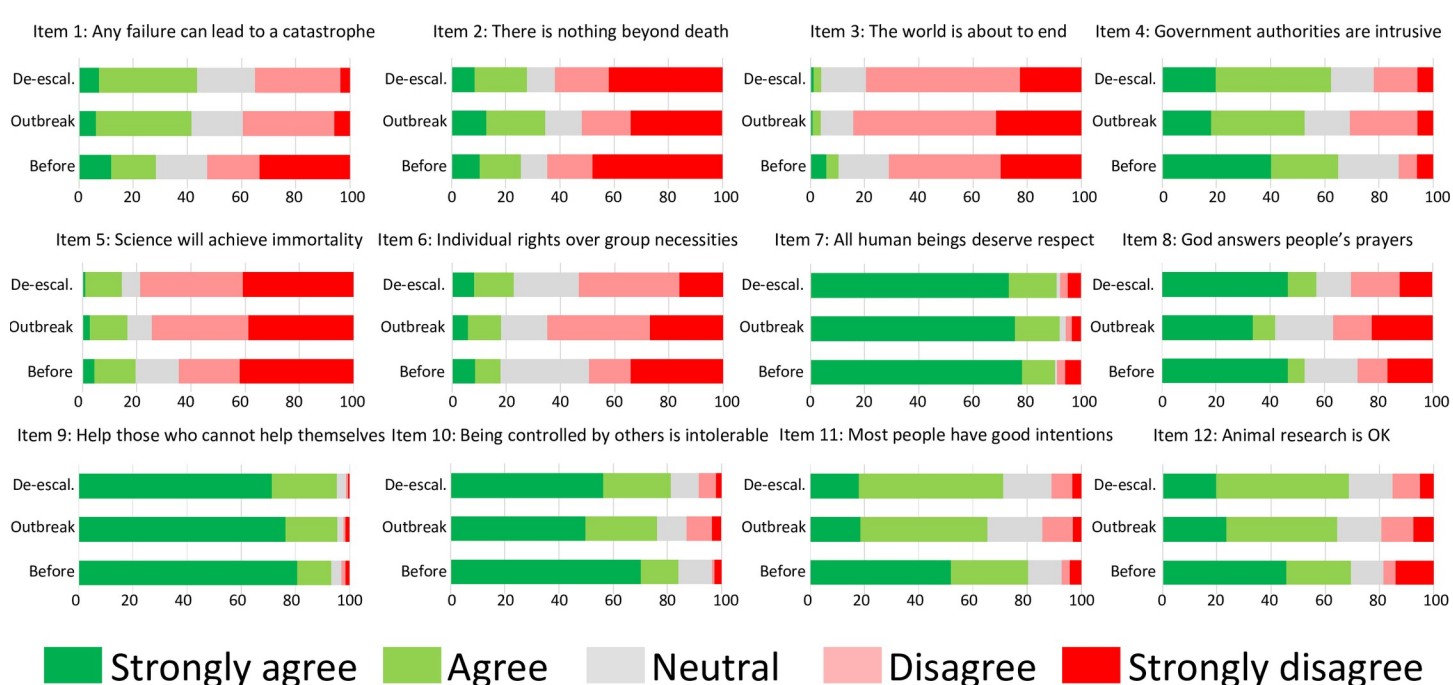

**Fig 1. Effect of the pandemic on personal beliefs.** Stacked bars graphic showing the percentage of participants that showed their strong (1) agreement (2), strong (5) disagreement (4), or neutrality (3) with every proposition before the pandemic, during the outbreak and de-escalation.

(0 = before COVID-19; 1 = outbreak; 2 = de-escalation) as main predictor, as well as 'politics' (0 = right, 1 = left, 2 = other, including ambiguous voters and participants who usually did not vote). We also added sex, age, civil status, COVID-19 sick acquaintance and COVID-19 deceased relative as covariates. Hence, the effect of each independent variable was controlled for the effects of the remaining covariates. Statistical results showing the effects of wave are summarized in Table 2, and information about the remaining covariates are detailed in S2 Table. We also present the 'average disagreement level' for each item and wave, which was computed for every item as the sum of each disagreement level (1 to 5) multiplied by the proportion of participants that responded that level of disagreement (S1 Fig).

The outbreak of the pandemic led respondents to agree more firmly with the proposition that any failure may lead to a catastrophe (item 1), and also with item 2 (there is nothing beyond death). However, it yielded a stronger disagreement with the world being about to end (item 3), authorities being intrusive (item 4), God answering people's prayers (item 8), the intolerability of being controlled by others (item 10), most people having good intentions (item 11) and animal research being okay (item 12). On the other hand, de-escalation (with respect to the outbreak of the pandemic) was associated with a significant stronger agreement with item 3 (the world is about to end), item 6 (individual rights are more important than group necessities), item 8 (God answers people's prayers) and item 10 (being controlled by others is intolerable). Conversely, respondents disagreed more strongly with the idea of there being nothing beyond death (item 2) in the de-escalation with respect to the outbreak, at a marginal level (p = 0.050).

These changes in the endorsement of the 12 propositions throughout the pandemic is further explored in S1 Text.

## Effect of political preferences on pandemic-related belief changes

Next, we asked whether political preference modified the impact of the pandemic on personal beliefs. We performed ordinal logistic regressions similar to those of the previous section, but including the interaction between 'wave' and 'politics'. A significant interaction was found for items 2, 3, 4, 5, 7 and 10 (Table 3 and Fig 2).

The percentage of participants that showed each disagreement level at each wave is summarized in S3 Table. In detail, left-sided voters were more affected by the pandemic on their opinion about there being nothing beyond death (item 2): the proportion of participants that strongly agreed with this proposition was higher after the outbreak and de-escalation (before: 12%; outbreak: 20%; de-escalation: 18.6%), and the proportion of left-sided voters that strongly disagreed with the proposition decreased with the pandemic (before: 44%; outbreak: 13.9%; de-escalation: 12.7%). With respect to item 3 (the world is about to end), the main effect was a stronger disagreement of left-sided voters during de-escalation: whereas opinions against this proposition decreased for right-sided voters from to 84.7% to 75.8%, it slightly increased for left-sided voters from 85.3% to 88.1%.

Interestingly, the endorsement of the pandemic-related change of agreement with the proposition about government authorities being intrusive clearly depended on political ideology. Left-sided voters overwhelmingly agreed with this proposition before the outbreak, but their opinion changed after the pandemic (before: 84% of respondents agreed or strongly agreed with the item; outbreak: 45.7%; de-escalation: 44.9%). However, this trend was the opposite for right-sided voters, especially in the de-escalation (before: 61.6%; outbreak: 57.6%; de-escalation: 70.4%), and for ambiguous voters (before: 55%; outbreak: 53.6%; de-escalation: 61.3%). In addition, left-sided voters became less-pessimistic after the de-escalation about science being able to achieve immortality (responses against the proposition: right-sided voters, 77.6%

**Table 2. Statistical data of the ordinal logistic regressions to assess the influence of the pandemic on each item.**

| Item 1 | | Any failure can lead to a catastrophe | | | | |
|---|---|---|---|---|---|---|
| | Model | LR $\chi^2$(15) = 58.24, p<0.0001, pseudo-R$^2$ = 0.0125 | | | | |
| | | OR | SE | 95% CI | z | p |
| | Before COVID (vs outbreak) | 3.42 | 0.74 | 2.23,5.23 | 5.68 | <0.001 |
| | De-escalation (vs outbreak) | 0.93 | 0.10 | 0.75,1.15 | -0.67 | 0.501 |
| Item 2 | | There is nothing beyond death | | | | |
| | Model | LR $\chi^2$(15) = 354.69, p<0.0001, pseudo-R$^2$ = 0.0710 | | | | |
| | | OR | SE | 95% CI | z | p |
| | Before COVID (vs outbreak) | 1.94 | 0.39 | 1.30,2.87 | 3.29 | 0.001 |
| | De-escalation (vs outbreak) | 1.24 | 0.14 | 0.99,1.55 | 1.96 | 0.050 |
| Item 3 | | The world is about to end | | | | |
| | Model | LR $\chi^2$(15) = 28.70, p = 0.0176, pseudo-R$^2$ = 0.0077 | | | | |
| | | OR | SE | 95% CI | z | p |
| | Before COVID (vs outbreak) | 0.63 | 0.13 | 0.42,0.94 | -2.24 | 0.025 |
| | De-escalation (vs outbreak) | 0.72 | 0.08 | 0.58,0.91 | -2.79 | 0.005 |
| Item 4 | | Government authorities tend to be intrusive and controlling | | | | |
| | Model | LR $\chi^2$(15) = 86.72, p<0.0001, pseudo-R$^2$ = 0.0177 | | | | |
| | | OR | SE | 95% CI | z | p |
| | Before COVID (vs outbreak) | 0.37 | 0.07 | 0.25,0.55 | -5.03 | <0.001 |
| | De-escalation (vs outbreak) | 0.86 | 0.09 | 0.69,1.06 | -1.44 | 0.151 |
| Item 5 | | Scientific progress can help us overcome death and live forever | | | | |
| | Model | LR $\chi^2$(15) = 133.89, p<0.0001, pseudo-R$^2$ = 0.0310 | | | | |
| | | OR | SE | 95% CI | z | p |
| | Before COVID (vs outbreak) | 1.20 | 0.24 | 0.82,1.76 | 0.94 | 0.350 |
| | De-escalation (vs outbreak) | 1.05 | 0.12 | 0.84,1.30 | 0.42 | 0.674 |
| Item 6 | | Individual rights are more important than the needs of any group | | | | |
| | Model | LR $\chi^2$(15) = 70.23, p<0.0001, pseudo-R$^2$ = 0.0144 | | | | |
| | | OR | SE | 95% CI | z | p |
| | Before COVID (vs outbreak) | 0.98 | 0.19 | 0.66,1.44 | -0.11 | 0.915 |
| | De-escalation (vs outbreak) | 0.64 | 0.07 | 0.52,0.79 | -4.08 | <0.001 |
| Item 7 | | All human beings deserve respect | | | | |
| | Model | LR $\chi^2$(15) = 67.33, p<0.0001, pseudo-R$^2$ = 0.0255 | | | | |
| | | OR | SE | 95% CI | z | p |
| | Before COVID (vs outbreak) | 0.63 | 0.17 | 0.38,1.06 | -1.75 | 0.080 |
| | De-escalation (vs outbreak) | 1.01 | 0.14 | 0.78,1.64 | 0.08 | 0.939 |
| Item 8 | | God answers people's prayers | | | | |
| | Model | LR $\chi^2$(15) = 539.41, p<0.0001, pseudo-R$^2$ = 0.1090 | | | | |
| | | OR | SE | 95% CI | z | p |
| | Before COVID (vs outbreak) | 0.62 | 0.12 | 0.42,0.91 | -2.41 | 0.016 |
| | De-escalation (vs outbreak) | 0.69 | 0.08 | 0.55,0.86 | -3.24 | 0.001 |
| Item 9 | | One should help those who are weak and cannot help themselves | | | | |
| | Model | LR $\chi^2$(15) = 21.38, p = 0.1253, pseudo-R$^2$ = 0.0090 | | | | |
| | | OR | SE | 95% CI | z | p |
| | Before COVID (vs outbreak) | 0.64 | 0.17 | 0.38,1.08 | -1.67 | 0.096 |
| | De-escalation (vs outbreak) | 1.19 | 0.16 | 0.92,1.56 | 1.31 | 0.190 |
| Item 10 | | Being controlled or dominated by others is intolerable | | | | |
| | Model | LR $\chi^2$(15) = 63.77, p<0.0001, pseudo-R$^2$ = 0.0157 | | | | |
| | | OR | SE | 95% CI | z | p |

(*Continued*)

**Table 2.** (Continued)

| | | | | | | |
|---|---|---|---|---|---|---|
| | Before COVID (vs outbreak) | 0.39 | 0.09 | 0.26,0.61 | -4.23 | <0.001 |
| | De-escalation (vs outbreak) | 0.76 | 0.09 | 0.60,0.95 | -2.38 | 0.017 |
| **Item 11** | | **Most people generally have good intentions** | | | | |
| | Model | LR $\chi^2(15)$ = 153.99, p<0.0001, pseudo-$R^2$ = 0.0350 | | | | |
| | | OR | SE | 95% CI | z | p |
| | Before COVID (vs outbreak) | 0.19 | 0.04 | 0.13,0.29 | -7.75 | <0.001 |
| | De-escalation (vs outbreak) | 1.01 | 0.11 | 0.81,1.27 | 0.12 | 0.901 |
| **Item 12** | | **It is okay to use animals for medical research** | | | | |
| | Model | LR $\chi^2(15)$ = 301.78, p<0.0001, pseudo-$R^2$ = 0.0640 | | | | |
| | | OR | SE | 95% CI | z | p |
| | Before COVID (vs outbreak) | 0.46 | 0.10 | 0.31,0.70 | -3.66 | <0.001 |
| | De-escalation (vs outbreak) | 1.11 | 0.12 | 0.90,1.38 | 0.97 | 0.332 |

Each model included item response (1 = strong agreement... 5 = strong disagreement) as dependent variable, wave and politics as predictors, and sex, age, civil status, COVID-19 sick acquaintance and COVID-19 deceased relative as covariates. Note that positive values of z and OR greater than 1 indicate a stronger disagreement with the proposition. Number of observations = 1650. Information for covariates are shown in S2 Table. OR, odds ratio; SE, standard error.

in the outbreak, 83.9% in the de-escalation; left-sided voters, 72.4% in the outbreak, 70.3% in the de-escalation).

With respect to all human beings deserving respect (item 7), the main difference was a significant decrease in a strong agreement with the proposition for left-sided and ambiguous voters during the outbreak of the pandemic (before: right-sided, 75.4%; left-sided: 94%; other, 80%; outbreak: right-sided, 80.7%; left-sided, 70.2%; other, 70.8%; de-escalation: right-sided, 76.3%; left-sided, 68.6%; other, 69.4%). Finally, left-sided voters agreed more strongly with the intolerability of being controlled by others before the pandemic, but this view changed during the pandemic (before: 88% strongly agreed; outbreak, 50%; de-escalation, 55.9%). This trend was similar for ambiguous voters (70%, 51.2% and 45.2%, respectively), but significantly different for right-sided voters, especially in the de-escalation (63.1%, 49.4% and 59.4%).

## Differential beliefs of participants with a COVID-19-affected acquaintance

Next, we asked whether beliefs of participants who reported having an acquaintance affected of COVID-19 were different to those who did not. To answer this, we performed ordinal logistic regressions as before, but excluding data before the pandemic. Once again, the response to each item was considered as dependent variable, reporting a COVID-19 sick acquaintance was the predictor of interest (1 = yes, N = 645; 0 = no, N = 901). Wave, sex, age, politics, civil status and reporting a COVID-19 deceased relative were included as covariates. Statistical results are summarized in S4 Table. Those respondents with an affected acquaintance disagreed more strongly with item 2 (there is nothing beyond death: 43.1% of respondents with an affected acquaintance answered 5, whereas 31.2% of the remaining participants did so), and item 5 (the capacity of science to achieve immortality: in the group with affected acquaintances, 12.9% agreed/strongly agreed and 78.8% disagreed/strongly disagreed, whereas figures were 18.2% and 73.4%, respectively, for the other group). However, they believed more firmly that government authorities are intrusive (item 4: 59.2% of participants with affected acquaintances agreed or strongly agreed with the proposition, versus 52.7% in the other group), all human beings deserve respect (item 7: 78.6% strongly agreed with the proposition in the group affected acquaintances vs 72.1%), God answers people's prayers (item 8: 44.3% vs 32.3%), people should help others in need (item 9: 78.3% vs 72.6%) and animal research is okay (item 12:

**Table 3. Change in the predicted probability of strongly agreeing or strongly disagreeing throughout the pandemic depending on political ideology.**

| | | Strongly agree | | | Strongly disagree | | |
|---|---|---|---|---|---|---|---|
| | | Contrast* | $\chi^2$ | p | Contrast | $\chi^2$ | p |
| **Item 2: There is nothing beyond death** | | | | | | | |
| Outbreak vs Before | | | | | | | |
| | Right | 0.013 | 1.86 | 0.1722 | -0.075 | 1.57 | 0.2109 |
| | Left | 0.156 | 31.65 | <0.0001 | -0.274 | 8.97 | 0.0027 |
| | Other | 0.075 | 6.23 | 0.0125 | -0.19 | 3.14 | 0.0764 |
| De-escalation vs Before | | | | | | | |
| | Right | -0.004 | 0.14 | 0.7047 | 0.025 | 0.15 | 0.6980 |
| | Left | 0.177 | 21.68 | <0.0001 | -0.290 | 9.68 | 0.0019 |
| | Other | 0.050 | 2.31 | 0.1283 | -0.149 | 1.68 | 0.1954 |
| **Item 3: The world is about to end** | | | | | | | |
| Outbreak vs Before | | | | | | | |
| | Right | -0.008 | 3.03 | 0.0819 | 0.126 | 7.45 | 0.0064 |
| | Left | -0.002 | 0.10 | 0.7489 | 0.028 | 0.12 | 0.7252 |
| | Other | -0.006 | 0.53 | 0.4680 | 0.073 | 0.91 | 0.3404 |
| De-escalation vs Before | | | | | | | |
| | Right | -0.001 | 0.08 | 0.7815 | 0.013 | 0.08 | 0.7730 |
| | Left | -0.001 | 0.08 | 0.7730 | 0.027 | 0.10 | 0.7573 |
| | Other | -0.005 | 0.31 | 0.5769 | 0.054 | 0.42 | 0.5155 |
| **Item 4: Government authorities tend to be intrusive and controlling** | | | | | | | |
| Outbreak vs Before | | | | | | | |
| | Right | -0.131 | 6.37 | 0.0116 | 0.026 | 11.46 | 0.0007 |
| | Left | -0.419 | 20.07 | <0.0001 | 0.066 | 50.82 | <0.0001 |
| | Other | -0.123 | 1.52 | 0.2173 | 0.026 | 3.04 | 0.0812 |
| De-escalation vs Before | | | | | | | |
| | Right | -0.057 | 1.08 | 0.2976 | 0.008 | 1.31 | 0.2520 |
| | Left | -0.452 | 23.02 | <0.0001 | 0.090 | 27.79 | <0.0001 |
| | Other | -0.116 | 1.28 | 0.2571 | 0.024 | 2.02 | 0.1548 |
| **Item 5: Scientific progress can help us overcome death and live forever** | | | | | | | |
| Outbreak vs Before | | | | | | | |
| | Right | 0.002 | 0.20 | 0.6520 | -0.027 | 0.19 | 0.6648 |
| | Left | 0.009 | 1.11 | 0.2931 | -0.079 | 0.78 | 0.3774 |
| | Other | 0.008 | 0.61 | 0.43432 | -0.065 | 0.47 | 0.4943 |
| De-escalation vs Before | | | | | | | |
| | Right | -0.002 | 0.37 | 0.5424 | 0.042 | 0.42 | 0.5193 |
| | Left | 0.020 | 3.44 | 0.0636 | -0.141 | 2.33 | 0.1268 |
| | Other | 0.008 | 0.49 | 0.4853 | -0.066 | 0.41 | 0.5201 |
| **Item 7: All human beings deserve respect** | | | | | | | |
| | | Contrast* | $\chi^2$ | p | Contrast | $\chi^2$ | p |
| Outbreak vs Before | | | | | | | |

*(Continued)*

**Table 3.** (Continued)

| | | Contrast* | χ² | p | Contrast | χ² | p |
|---|---|---|---|---|---|---|---|
| | Right | 0.022 | 0.18 | 0.6740 | -0.004 | 0.17 | 0.6804 |
| | Left | -0.262 | 41.84 | <0.0001 | 0.044 | 29.60 | <0.0001 |
| | Other | -0.118 | 1.66 | 0.1974 | 0.024 | 1.96 | 0.1620 |
| De-escalation vs Before | | | | | | | |
| | Right | 0.003 | 0.00 | 0.9596 | -0.001 | 0.00 | 0.9597 |
| | Left | -0.253 | 23.63 | 23.63 | 0.042 | 15.65 | 0.0001 |
| | Other | -0.088 | 0.74 | 0.74 | 0.017 | 0.83 | 0.3629 |
| **Item 10: Being controlled or dominated by others is intolerable** | | | | | | | |
| | | Strongly agree | | | Strongly disagree | | |
| | | Contrast* | χ² | p | Contrast | χ² | p |
| Outbreak vs Before | | | | | | | |
| | Right | -0.151 | 6.00 | 0.0143 | 0.015 | 7.41 | 0.0065 |
| | Left | -0.375 | 32.38 | <0.0001 | 0.026 | 26.27 | <0.0001 |
| | Other | -0.216 | 4.33 | 0.0374 | 0.019 | 5.79 | 0.0162 |
| De-escalation vs Before | | | | | | | |
| | Right | -0.052 | 0.65 | 0.4197 | 0.004 | 0.70 | 0.4024 |
| | Left | -0.325 | 18.41 | <0.0001 | 0.021 | 12.80 | 0.0003 |
| | Other | -0.229 | 4.18 | 0.0409 | 0.021 | 4.37 | 0.0365 |

Only those items with a significant interaction between wave and politics are shown.

*"Contrast" refers to the change in the predicted probability to respond 1 (strongly agree) or 5 (strongly disagree) for each ideology group. For example, for left-sided voters, the probability of strongly agreeing with item 2 in the outbreak was 15.6% higher than before the COVID-19. However, this increase was much lower (1.3%) for right-sided voters.

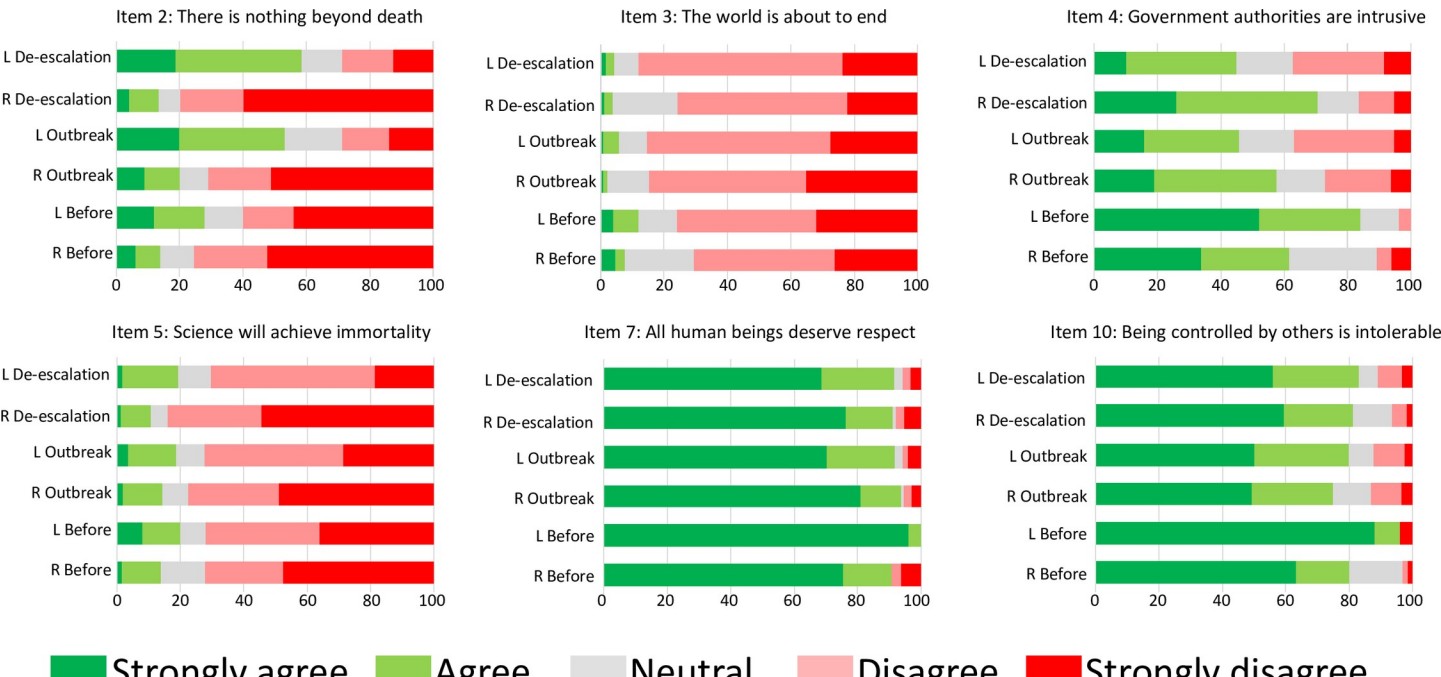

**Fig 2. Impact of political preference on the effect of the pandemic on beliefs.** Stacked bars graphic showing the percentage of participants that showed their strong (1) agreement (2), strong (5) disagreement (4), or neutrality (3) with every proposition before the pandemic, during the outbreak and de-escalation, stratified by political preference (right-sided and left-sided voters; ambiguous voters not shown).

28.1% vs 18.4%). With regards to a differential belief change throughout the pandemic depending on reporting an affected acquaintance, ordinal regressions including the interaction between affected acquaintance and wave did not yield significant results.

Finally, we repeated the same analyses for participants reporting a COVID-19 deceased relative, restricting the observations to the de-escalation (since this question was not included in the survey at the outbreak, when there were few confirmed casualties in Spain). These ordinal logistic regressions included 37 participants with a deceased relative, and 404 in the remaining group. Once again, sex, civil status, age, political preference and having a sick acquaintance were included as covariates. In this case, none of the analyses reached statistical significance, although there was a marginal effect for item 4 (p = 0.062): respondents who reported a deceased relative tended to believe more strongly that authorities are intrusive (results are shown in S5 Table).

### Replication of results on weighted data after iterative proportional fitting

Since the samples recruited in the three waves were not representative of the total Spanish population in terms of sex, age and political preference, we used iterative proportional fitting (i.e. raking) to assign a weight to each volunteer in order to correct under or overrepresentation with respect to their sociodemographic group (combining sex and age group) and political preference. Thus, after raking, we carried out ordinal logistic regressions as explained above. Full outputs are uploaded as Supporting information together with this manuscript (S1 File); also, the effect of the pandemic and the influence of political preference is described in this section, and illustrated in S2 Fig, where stacked bars obtained with raw and weighted data are compared.

In the previous section, we showed that items 2, 3, 4, 5, 7 and 10 had a significant interaction between wave and political preference. After considering data weights, the interaction for items 3, 4, 7 remained significant, as well as for item 1. In detail, the proportion of right-sided voters that strongly agreed with any failure can lead to a catastrophe (item 1) in the outbreak increased with respect to before the pandemic (2.74% vs 8.49%). However, it followed an opposite trend for left-sided voters (14.49% vs 5.44%). Similarly, the probability of strongly disagreeing with this item for a right-sided voter was on average about 51% higher in the de-escalation than before the pandemic, whereas changes for left-sided voters were subtler (7.6% increase).

About item 3 (The world is about to end), the most significant result was the increased proportion of right-sided voters that strongly disagreed with it in the outbreak compared with before the pandemic: on average, the proportion of this subsample of participants that answered 5 to this item increased from 13.5 to 35%, whereas it decreased (from 32.8 to 24.4%) for left-sided voters. Note that this trend was similar to the analyses on raw data.

As expected, weighted data also revealed differences for item 4 (Government authorities are intrusive): whereas the probability for a right-sided voter to strongly agree and strongly disagree with this item was unchanged in the de-escalation (with respect to before the pandemic), the probability for a left-sided voter to strongly agree decreased by 47%, and the probability to strongly disagree increased by 10.5%.

Finally, the effect on item 7 (All human beings deserve respect) was also replicated. The main contributor to this interaction was a decrease in the probability of left-sided voters to strongly agree with this item (95.27% before the pandemic vs 68.71% after the outbreak, being 66.33% in the de-escalation). Figures were more stable for right-sided voters (before COVID-19: 80.40%; outbreak: 80.31%; de-escalation: 74.02%). In conclusion, analyses on weighted data mostly replicated those performed on raw data.

With respect to differential beliefs of those participants reporting a COVID-19 affected acquaintance (waves 2 and 3), significant results were found for items 4, 5, 8, 11 and 12 (versus 2, 5, 7, 8 and 12 on raw data). In detail, 63.8% of participants with an affected acquaintance agreed or strongly agreed with government authorities being intrusive, whereas this proportion was lower (53.7%) for the remaining participants. However, this subset of participants was more pessimistic about science achieving immortality: 43.1% strongly disagreed (vs 30.5% of the remaining participants) with item 5. Following the same trend as for the raw data, 41.7% of participants with an affected relative strongly believed that God answers people's prayers (item 8), whereas this proportion was lower (28.9%) for the rest. Similarly, a higher proportion of the former (77.3%) agreed/strongly agreed with item 11 (Most people have good intentions), versus 67.8% in the other group. Finally, analyses on weighted data also showed an increased agreement/strong agreement in participants with an affected relative with animal experimentation (72.11% vs 62.02% for the remaining participants), as well as a decreased disagreement/ strong disagreement (12.03% vs 21.98%).

Analyses on weighted data were more sensitive to detect differential beliefs of participants with a deceased relative during de-escalation: item 11 showed significant differences (z = -2.10, p = 0.036), and item 10 showed a marginal effect (z = -1.96, p = 0.05). Interestingly, according to these analyses, a higher proportion of participants with a deceased relative strongly agreed with item 11 (Most people have good intentions): 36.5% vs 17.7% for the remaining participants. With respect to item 10 (Being controlled by others is intolerable), the main difference is that 75.77% of participants with a deceased relative strongly agreed with the proposition, compared with 53.31% of the remaining volunteers.

## Pandemic and political preference: Longitudinal data

Besides replicating the main results of the study by weighting data with iterative proportional fitting, we performed the analyses on a subsample of longitudinal data, that is, participants who were evaluated at waves 2 and 3. Based on the anonymous code provided by respondents, as well as their sociodemographic information, we identified 97 participants (sex: 52 female; age: 33 18–31 yr, 18 31–40 yr, 27 41–50 yr, 10 51–60 yr, 9 60+ yr; civil status: 42 single; 51 married; 1 widowed; 1 divorced; 2 domestic partner; political preference: 60 right-side voters; 25 left-side voters; 12 'other') who completed the assessment in the outbreak and de-escalation time points. To corroborate our previous results, we ran ordinal logistic mixed models for each item to ask whether there was a differential impact of the pandemic on beliefs, depending on political preference. Since sample size was relatively small, we merged data on political preference in a variable termed 'right voter' (0 = no, including left-sided voters and 'other' political preference, N = 37; 1 = yes, N = 60). Age was also recoded as an 'older' variable (0 = 18–40 yr, N = 51; 1 = older than 40, N = 46), and civil status was transformed to 'single' (0 = married and domestic partners, N = 53; 1 = single, N = 42). Thus, the ordinal mixed models included the response to each item as dependent variable, a fixed-effects equation with a factorial interaction between wave and 'right-sided voter', and sex, 'older', 'single', COVID-19 sick acquaintance and COVID-19 deceased relative as covariates, and a random-effects equation with wave nested within subjects.

The multilevel models revealed a significant interaction between wave and political preference for item 4 (model significance: $\chi^2(8)$ = 22.51, p = 0.0041), and for item 6 (model significance: $\chi^2(8)$ = 17.70, p = 0.0236) (statistical data are shown in S6 Table). The change in predicted probabilities for significant items is summarized in Table 4. Also, the percentage of respondents that expressed each disagreement level for every item is shown in S1 File, sorted by political ideology (right-sided voter, yes/no) and wave (outbreak/de-escalation).

**Table 4. Change in the predicted probability of strongly agreeing or strongly disagreeing throughout the pandemic depending on political ideology, based on the longitudinal data (N = 97, repeated measures).**

| | | Strongly agree | | | Strongly disagree | | |
|---|---|---|---|---|---|---|---|
| Item 4: Government authorities tend to be intrusive and controlling | | | | | | | |
| | | Contrast* | $\chi^2$ | p | Contrast | $\chi^2$ | p |
| De-escalation vs Outbreak | | | | | | | |
| Right-sided voter | | 0.193 | 12.37 | 0.0004 | -0.052 | 5.58 | 0.0181 |
| No right-sided voter | | -0.124 | 2.60 | 0.1066 | 0.021 | 1.94 | 0.1635 |
| Item 6: Individual rights are more important than the needs of any group | | | | | | | |
| | | Strongly agree | | | Strongly disagree | | |
| | | Contrast* | $\chi^2$ | p | Contrast | $\chi^2$ | p |
| De-escalation vs Outbreak | | | | | | | |
| Right-sided voter | | 0.041 | 5.18 | 0.0228 | -0.176 | 12.59 | 0.0004 |
| No right-sided voter | | 0.007 | 0.14 | 0.7098 | -0.020 | 0.14 | 0.7073 |

Results are based on ordinal logistic mixed models to predict level of disagreement with each item. Only those items with a significant interaction between wave and politics are shown. The fixed-factor equation is composed of a full interaction between wave and politics as predictors, and sex, older (= 0 if 18–40 yr, 1 = if >41 yr), single (= 0 if married/domestic partner; = 1 if single), COVID-19 sick acquaintance (1 = yes, 0 = no) and COVID-19 deceased relative (1 = yes, 0 = no) as covariates. Random effects: wave nested within subjects. Random effects were significant only for item 6 ($\chi^2(1)$ = 7.95, p = 0.0024). Note that positive values of z and OR greater than 1 indicate a stronger disagreement with the proposition. OR, odds ratio; SE, standard error.

*"Contrast" refers to the change in the predicted probability to respond 1 (strongly agree) or 5 (strongly disagree) for each political group. For example, for right-sided voters, the probability of strongly agreeing with item 4 in the de-escalation was 19.3% higher than in the outbreak. However, the predicted probability for left-sided and ambiguous voters to strongly agree with this proposition decreased 12.4% throughout the pandemic.

With regards to these data, right-sided voters significantly changed their mind from the outbreak to the de-escalation on the idea of authorities being intrusive: in the outbreak, 40% of right-sided respondents agreed with the proposition, and the same proportion was against it; during the de-escalation, these figures were 78.3% and 11.7% for and against, respectively. This effect was not found for non-right-sided voters (outbreak: 70.3% for and 18.9% against the proposition; de-escalation, 51% for and 21.6% against the proposition). This effect was also found in the predicted probabilities of the multilevel model shown in Table 4: for right-sided voters, there was a 19.3% increased probability of strongly agreeing with authorities being intrusive in the de-escalation with respect to the outbreak; however, for non-right-sided voters, there was a 12.3% decreased probability of strongly agreeing in the de-escalation that in the outbreak. With regards to item 6 (preeminence of individual rights over group necessities), there was a marginal effect (p = 0.054): based on percentages shown in S1 File, 12% of right-sided voters agreed with the proposition at the outbreak, and this figure increased to 25% in the de-escalation. However, it remained nearly unchanged for non-right-sided voters (27% vs 24.3%). As it is shown in Table 4, this effect was also present as a decreased predicted probability (17.6%) of right-sided voters to strongly disagree with this item in the de-escalation with respect to the outbreak. By contrary, this decrease for non-right-sided voters was just 2%.

Finally, we computed how individual opinions changed through time by subtracting responses in the de-escalation from those in the outbreak. Positive values were coded as "increased agreement", negative values as "decreased agreement", and zeroes as "unchanged agreement". These changes in level of agreement are displayed in S3 Fig, stratified by political preference. Opinions remained mostly unchanged for items 7 and 9, for example, which refer to basic human rights. However, the effect of politics on items 4, 6, 8 or 10 shown previously is also clear in this case, where individual changes were assessed.

## Discussion

We show that the COVID-19 pandemic had a significant impact on social, interpersonal and transcendental beliefs of Spanish residents. This goes in line with previous research suggesting that extraordinary and traumatic events may change personal beliefs [22]. The main novelty of this research is to assess belief changes due to the COVID-19 pandemic across several domains, which is especially valuable for having been carried out in one of the first and most affected countries.

Interestingly, despite the restrictive measures taken by the Spanish government during the outbreak, respondents were permissive with these restrictions and tolerated being controlled by others. This effect was also found during the H1N1 epidemic: Bangerter and collaborators [6] demonstrated that trust in political institutions increased throughout the crisis. Our study supports these findings, but also shows that the effect reversed during de-escalation: after several weeks of total lockdown, authorities were considered excessively intrusive and agreement on the intolerability of being controlled boosted. With regards to interpersonal beliefs, we detected a significant mistrust in other people's intentions in the outbreak, and a strengthened individualistic attitude during de-escalation. Both findings support a detachment of social bounds as a consequence of the pandemic [2]. The former might be due to the plethora of videos of citizens violating the lockdown, which circulated in mass media and social networks; the latter has already been suggested by other reports on the effects of the COVID-19 on economic individualistic behaviors [23]. On the other hand, transcendental beliefs experienced a non-significant decrease during the outbreak, but were significantly strengthened in the de-escalation. Preliminary research on the effect of the COVID-19 pandemic on religious beliefs of US and UK citizens points to a polarizing effect: strong believers reported higher confidence in their beliefs, whereas non-believers declared a stronger skepticism on religion [24]. Previous reports on the role of traumatic events on religious beliefs reported mixed results [15, 16]. Our data contribute to this topic by showing that respondents with a COVID-19 affected acquaintance agreed more strongly with transcendental beliefs, such as the existence of an afterlife and God answering prayers. Besides, prosocial beliefs (i.e. all human beings deserve respect) were also strengthened, pointing to an enhancement of the affective components of belief systems [25]. Interestingly, the attitude towards science was more supportive, but realistic at the same time: support of animal research was stronger, although agreement with the capacity of science to achieve immortality was weaker than in the group without affected acquaintances.

In any case, the effect of the pandemic on personal beliefs is strongly modulated by political preference. The most remarkable case is the belief in authorities being intrusive, especially during de-escalation: whereas a majority of right-sided and ambiguous voters endorsed this proposition, support by left-sided voters was below 50%. It is important to note that, during the COVID-19 crisis, the Spanish Government was composed of a socialist-communist coalition. Thus, as suggested by previous reports [26] left-sided voters could be more permissive with their decisions, and right-sided voters could more easily react against them. Nevertheless, this effect seems to be specific of the crisis, because before the COVID-19 pandemic nearly 80% of left-sided voters agreed or strongly agreed with authorities being intrusive, even though the socialist party was already in the government. This result was solidly replicated by the longitudinal dataset: even though previous reports suggest that belief change in adulthood is marginal [27, 28], our results show that it could be facilitated by extraordinary events.

In line with these results, during de-escalation, the proportion of left-sided voters that found tolerable being controlled by others remained strikingly higher than before the pandemic, whereas for right-sided voters values were restored to pre-pandemic levels of disagreement. Regarding transcendental beliefs, they were attenuated in left-sided voters, since the

proportion of respondents that agreed with there being nothing beyond death increased with the pandemic. All these politically-dependent results point to a polarization as a consequence of the COVID-19 crisis. In recent years, there has been an increasing interest in belief polarization studies [29–32], pinpointing the importance of the Internet and social media on this process [33]. A 'belief polarization' occurs when two people respond to the same evidence by updating their beliefs in opposite directions [34]. There are studies for [35, 36] and against [37–40] this phenomenon. Our results suggest that political-preference groups are more polarized in terms of transcendental and social beliefs as a consequence of the pandemic.

Our research has a set of limitations. First, although we present a subset of longitudinal analyses, most of the data were cross-sectional. Therefore, 'belief change' does not refer to a within-subject modification, but a difference between three 'snapshots' describing beliefs at a population level. Second, the sample assessed before the pandemic was smaller than in the other time points; hence, it could be argued that those results are not representative of the general population. In any case, sample size in the outbreak and de-escalation was similar to previous reports, so belief changes during the different stages of the pandemic are expected to be solid. Besides, we also show similar results when data was weighted to correct for over or underrepresentation of sociodemographic and political preference groups. Third, as it is intrinsic in any survey-based research, participants might understand the propositions and response options in various ways. Whereas some beliefs are susceptible to be consciously expressed [41], others may be more difficult to be made explicit [42]. Besides, we assess the conscious endorsement of a set of propositions, but the actual influence on behavior of those propositions remains unknown. With regards to the formulation of items, one of them was negatively expressed (item 2: "There is nothing beyond death"). As it has been suggested by previous studies [43], participants might be more prone to disagree with a proposition expressed in negative terms. In fact, a study published by the Spanish Center for Sociological Research in 2018 [44] (see here: https://bit.ly/3o4dHyE), shows in question 44 (positively formulated: "Do you believe in life after death?") that 17.9% of participants responded "absolutely yes", whereas 29.6% responded "absolutely not". These numbers are at odds with our results. However, this would not have an impact on the main goal of our research, which is the effect of the pandemic on each individual item. Since all items were formulated in the same way across all three waves, positive or negative formulation should not alter the impact of the pandemic on its endorsement.

Because the SARS-CoV-2 is a highly contagious disease with human-to-human transmission, our ability to minimize the consequences of the COVID-19 pandemic depends, to a great extent, on the coordinated response of political authorities and the citizenry. To the extent that beliefs provide the foundation for attitudes and behaviors, understanding the ways in which the pandemic might be affecting beliefs about the legitimacy of the government, the trustworthiness of others, scientific progress or the preeminence of the individual over the collective, might be crucial for increasing acceptance and adherence to health guidelines [45]. This should be a priority for psychological science [18]. In turn, recent research shows that US citizens holding 'faith in Trump' (quantified by two specific scales, including items such as "President Trump will make America healthy again") are more reluctant to keep the recommended social distance [46]. Beliefs allow us to assess ordinary and extraordinary events within the wider context of our past and expectations for the future. They go beyond episodic memories [47], and have an impact on cognitive and emotional components that influence our behavior [42, 48].

In conclusion, we show that transcendental, social and interpersonal beliefs have changed in Spanish residents as a consequence of the COVID-19 pandemic. More importantly, these changes depend on political preference, and point to a polarization of society. Government

authorities should be aware of their influence on society, especially on their voters, and promote responsible behavior in accordance with international guidelines. Moreover, they should consider whether polarization, irrespective to electoral interests, is beneficial for societal wellbeing.

## Supporting information

**S1 Text. Supplementary methods, results and reference.** Full explanation of the iterative proportional fitting procedure to weight data. Supporting analyses on "Personal beliefs, political preference and sociodemographic groups", and "Effect of the pandemic on personal beliefs". Additional reference included in Supplementary Methods.
(DOCX)

**S1 Datasets. Main (cross-sectional) and longitudinal datasets used for this study.** Stata and csv files are provided. Weighted data (after iterative proportional fitting) are also included.
(ZIP)

**S1 File. Stata's log files in plain text format including all analyses used across the manuscript.**
(ZIP)

**S1 Table. Proportion of participants (%) that showed their strong (1) agreement (2), strong (5) disagreement (4), or neutrality (3) with every item (N = 1706).** Concerning answers, 1 = "I agree, and I would continue to agree even if I were shown 'irrefutable' proof to the contrary"; 2 = "I agree, although I could change my mind if I were shown strong evidence"; 3 = "I neither agree nor disagree"; 4 = "I disagree, although I could change my mind if I were shown strong evidence"; 5 = "I disagree, and I would continue to disagree even if I were shown 'irrefutable' proof".
(DOCX)

**S2 Table. Statistical data of the ordinal logistic regressions to assess the influence of politics and sex on each item, across all time points (N = 1706).** Each model included item response (1 = strong agreement. . . 5 = strong disagreement) as dependent variable, political preference and wave as predictors, and sex, age, civil status, COVID-19 sick acquaintance and COVID-19 deceased relative as covariates. Note that positive values of z and OR greater than 1 indicate a stronger disagreement with the proposition. Number of observations = 1650. OR, odds ratio; SE, standard error.
(DOCX)

**S3 Table. Proportion of participants (%) that showed their (strong = 1) agreement (= 2), (strong = 5) disagreement (= 4) or neutrality (= 3) with every item.** [a]Number of respondents: items 1, 2 and 8, N = 144; items 3 and 5, N = 138; items 4 and 6, N = 117; item 7, N = 156; items 9 and 12, N = 134; item 10, N = 114; item 11, N = 123. [b]Significant differences between before COVID-19 and outbreak (see Results for details) [c]Significant differences between outbreak and de-escalation (see Results for details) Main contributors to significant differences are in bold typeset. A critical value of 0.00027 (i.e. Bonferroni correction for 12 survey items and 15 cells in each contingency table: $0.05/(12{*}15) = 0.00027$) was selected; since adjusted residuals follow a normal distribution with mean = 0 and SD = 1, the critical value selected for adjusted residuals was 3.45. In conclusion, numbers in bold typeset point to those values whose adjusted residuals were greater than 3.45.
(DOCX)

**S4 Table. Statistical data of the ordinal logistic regressions to assess differential responses of participants with a COVID-19 sick acquaintance (= 1, yes; = 0, no), restricted to waves 'outbreak' and 'de-escalation'.** Results are restricted to outbreak and de-escalation. Each model included item response (1 = strong agreement. . . 5 = strong disagreement) as dependent variable, COVID-19 sick acquaintance as predictor (= 1, yes; = 0, no), and wave, politics, sex, age, civil status and COVID-19 deceased relative as covariates. Number of observations = 1540. Note that positive values of z and OR greater than 1 indicate a stronger disagreement with the proposition. OR, odds ratio; SE, standard error.
(DOCX)

**S5 Table. Statistical data of the ordinal logistic regressions to assess differential responses of participants with a COVID-19 deceased relative (= 1, yes; = 0, no), restricted to de-escalation.** Results are restricted to de-escalation (N = 441). Each model included item response (1 = strong agreement. . . 5 = strong disagreement) as dependent variable, COVID-19 deceased relative as predictor (= 1, yes; = 0, no), and politics, sex, age, civil status and COVID-19 sick acquaintance as covariates. Number of observations = 439. Note that positive values of z and OR greater than 1 indicate a stronger disagreement with the proposition. OR, odds ratio; SE, standard error.
(DOCX)

**S6 Table. Ordinal logistic mixed models for the longitudinal data (N = 97, two time points).** Each model included a fixed-effects and a random-effects equation. Fixed effects: item response (1 = strong agreement. . . 5 = strong disagreement) as dependent variable, full interaction between wave and politics as predictors, and sex, older (= 0 if 18–40 yr, 1 = if >41 yr), single (= 0 if married/domestic partner; = 1 if single), COVID-19 sick acquaintance (1 = yes, 0 = no) and COVID-19 deceased relative (1 = yes, 0 = no) as covariates. Random effects: wave nested within subjects. Random effects were significant only for item 6 ($\chi^2(1)$ = 7.95, p = 0.0024) Note that positive values of z and OR greater than 1 indicate a stronger disagreement with the proposition. OR, odds ratio; SE, standard error.
(DOCX)

**S1 Fig. Bar graphic showing the average disagreement level for each item and wave.** The value of each bar is calculated as the sum of the proportion of participants that responded each possible value (1 to 5) multiplied by that value. See S2 Table for a description of items. Note that higher values indicate a stronger disagreement with the proposition.
(TIF)

**S2 Fig. Effect of the pandemic and modulation of political preference on beliefs, comparing analyses on raw and weighted data.** Stacked bars graphic showing the proportion of participants that responded to each disagreement level (from "strongly agree" to "strongly disagree"), stratified by wave and political preference (right-sided and left-sided voters), both with raw and weighted data (after iterative proportional fitting; see Materials and Methods).
(TIF)

**S3 Fig. Individual changes in beliefs based on longitudinal data.** For each participant of the longitudinal dataset, responses to each item in the de-escalation were subtracted from those in the outbreak. Then, positive values were categorized as 'increased agreement', negative values as 'decreased agreement', and zeroes as 'unchanged agreement'. Histograms shows the percentage of participants that increased, decreased or did not change their agreement between both waves, stratified by political preference (left, no right-sided voter; right, right-sided voter).
(TIF)

## Author Contributions

**Conceptualization:** Javier Bernacer, Eduardo Camina, Francisco Güell.

**Data curation:** Javier Bernacer, Eduardo Camina.

**Formal analysis:** Javier Bernacer, Javier García-Manglano.

**Investigation:** Eduardo Camina, Francisco Güell.

**Methodology:** Javier García-Manglano.

**Supervision:** Francisco Güell.

**Visualization:** Javier Bernacer, Javier García-Manglano.

**Writing – original draft:** Javier Bernacer, Javier García-Manglano, Francisco Güell.

**Writing – review & editing:** Javier Bernacer, Javier García-Manglano, Eduardo Camina, Francisco Güell.

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
