## [Decision Letter · Decision Letter 0]

16 Feb 2021

PONE-D-20-35324

Impact of the COVID-19 pandemic on the belief system: the case of Spain

PLOS ONE

Dear Dr. Bernácer,

Thank you for submitting your manuscript to PLOS ONE. After careful consideration, we feel that it has merit but does not fully meet PLOS ONE’s publication criteria as it currently stands. Therefore, we invite you to submit a revised version of the manuscript that addresses the points raised during the review process.

Two reviewers have carefully read your paper and both are quite enthusiastic about the topic and importance of your timely analysis of attitudes before, during, and after the initial outbreak of coronavirus in Spain.  Both reviewers offer numerous specific suggestions for improvement of the paper, which I hope you will find helpful.  Please attend carefully to their suggestions. 

An overall concern that I share with the reviewers is that you find a way to focus your findings and find a more concise, coherent presentation of the results.  Reviewer 1 mentions the placement of the Methods section at the end, and I also found it necessary to read the end before I could understand the results and discussion section.  Another concern is with the sample: please expand on the purpose of the initial study, which used snowball sampling and network connections.  It seems that similar nonprobability sampling methods were used for the second and third waves.  This is likely the reason for the rather small proportion of older persons in the sample.  As one reviewer suggests, please compare the age structure of your sample with that of the nation.  

We look forward to receiving your revised manuscript.

Kind regards,

Ellen L. Idler

Academic Editor

PLOS ONE

Journal Requirements:

2.Thank you for stating the following in the Acknowledgments Section of your manuscript:

"This research has been supported by the Institute for Culture and Society (non-competitive funding)"

 "The authors received no specific funding for this work"

3. Please include a copy of Table 4 which you refer to in your text on page 21.

Reviewers' comments:

Reviewer's Responses to Questions

**Comments to the Author**

1. Is the manuscript technically sound, and do the data support the conclusions?

Reviewer #1: Partly

Reviewer #2: Yes

2. Has the statistical analysis been performed appropriately and rigorously? 

Reviewer #1: I Don't Know

Reviewer #2: Yes

3. Have the authors made all data underlying the findings in their manuscript fully available?

Reviewer #1: Yes

Reviewer #2: Yes

4. Is the manuscript presented in an intelligible fashion and written in standard English?

Reviewer #1: No

Reviewer #2: Yes

5. Review Comments to the Author

Reviewer #1: The authors conduct a three-wave survey do examine how the COVID19 pandemic and political affiliation affect a range of beliefs. They have a good grasp of psychological literature and an adequate knowledge of statistical analyses. However, the article suffers from disorganization and lack of focus. I provide detailed comments for each section below.

INTRO

Good intro and historical setup.

PREVIOUS RESEARCH

The authors should be clear that this is about “period effects” as distinct from other types of attitude change. H1N1 example is good, perhaps more detail on this as a prior example. The use of shattered assumptions theory is good, and the authors could also mention work in sociology on cultural trauma (Alexander) and natural disasters (Erikson). On the topic of religious coping, Kenneth Pargament’s work would be relevant here. The psych of conspiracy theories doesn’t seem relevant in this section, especially as it doesn’t come up much in remainder of paper. In line 116, “whether a set of beliefs changed,” is a weak research question. In line 120, “will encompass” is a vague hypothesis. Be more specific about hypotheses and expectations.

Methods section should come before results. I had to find / read this section before making sense of results.

SAMPLE

The authors make use of a pre-existing survey to approximate a longitudinal survey in three waves. However, the unbalanced sample sizes should be justified / addressed. Wave 1 is almost 1/10th the size of Wave 2 (156 vs. 1182). The addition of 97 linked respondents between W2 and W3 helps the argument.

ANALYSIS

The authors use ordinal logistic regression, to test effect of the pandemic, politics, and their interaction on personal beliefs. In line 504, "dependent" should be independent. In line 520-521, why present “differences in predicted probabilities”? These are not present in table 3 as mentioned, and table 4 is missing. In line 521 to 527, was sex not included in main analyses? This section was confusing.

RESULTS

The authors find a handful of significant relationships, but need to do a better job tying them together into a coherent story. This may mean ignoring some significant relationships to highlight substantively related findings. Tables S1-S4 are confusing. Why not present all covariates at once? No need to present four tables for one regression analysis. Does figure 1 come from Table S5? If so, the figure should depict significant differences somehow. Line 185-188 is unclear: chi-squared tests on what propositions? “analyzed residuals”? This procedure needs clarification.

Figure 2 is interesting, but there are many redundant lines. Is there a way to depict overall opinion rather than showing mirrored agree/disagree lines? ie. If fewer people agree with a statement, more will disagree. As-is this graph is hard to interpret. Table 2 is good.

DISCUSSION

The authors find that during the outbreak, more people accepted government control and skewed towards group priorities over individual priorities. During de-escalation, these feelings waned.

I recommend adding justifications / clarifications as described above, and streamlining the discussion of results to focus on main findings. Best of luck to the authors.

Reviewer #2: This work presents the result from a sequential survey conducted in Spain across three temporal waves: one year before the COVID-19 pandemic, at its outbreak, and during de-escalation. Respondents were asked to declare their level of agreement with 12 propositions concerning transcendental, social or interpersonal subjects. The aim of the authors was to investigate the effect of the pandemic on the beliefs related to the propositions. They also investigated whether the effect was influenced by some features of the respondents: age, gender, civil status, political ideology, having an acquaintance affected of COVID-19, and a COVID-19 deceased relative.

The authors claim that:

- Despite the lockdown, respondents tolerated being controlled by authorities, and acknowledged the importance of group necessities over individual rights.

- De-escalation changed the above beliefs.

- Transcendental beliefs were strengthened.

- Left-sided voters did not see authorities as intrusive

- Transcendental beliefs prevailed among right-sided voters.

- The pandemic resulted in a polarization of belief system based on political ideology.

GENERAL EVALUATION

The study presented here is of the highest relevance. On the one hand, psychologists and sociologists will gain knowledge about the impact of events like the COVID-19 pandemic on the belief system of individuals. On the other hand, politicians and international organizations may use results of this kind to modulate their policies according to their expected impact on the population. I would really like to see studies like this one performed in other countries and when facing other traumatic events.

The paper is written in a clear way, with an accurate yet easy to follow English. The authors have made a big effort to dissect the results of the survey and extract from them relevant and potentially beneficial information regarding the effect of the pandemic on beliefs. I think that the claims made in the abstract are well addressed, and they are well supported by the results of the survey.

However, I am going to express two concerns about the main claims made by the authors. I will discuss other minor concerns and suggestions that, if addressed, I think will enrich this already superb work.

MAIN CONCERNS

- In the abstract, the authors claim that ‘left-sided voters did not see authorities as intrusive’. But in light of Fig 2, item 4, and Table S6, I think it is more accurate to say that the proportion of left-sided voters that agreed with the proposition ‘Government authorities are intrusive’ decreased dramatically, and the proportion that disagreed had the opposite trend (which is a striking result by itself). But still the level of agreement was slightly higher (or at least not any lower) than the level of disagreement.

- Similarly, in lines 360 and 361 of the Discussion, the authors claim that ‘during de-escalation, left-sided voters found tolerable being controlled by others’. But in lines 225-228 of the Results, as well as in Fig 2, item 10, what we see is that the proportion of left voters that agreed with the proposition ‘Being controlled by others is intolerable’ decreased dramatically (which, again, is a remarkable result). But still the level of agreement with the intolerability of being controlled by others is much higher.

In the two above cases, I think that slight rephrasing of the sentences might be enough. For example:

• ‘a remarkably higher proportion of left-sided voters did not see authorities as intrusive after the outbreak’.

• ‘during de-escalation, the proportion of left-sided voters that found tolerable being controlled by others remained overwhelmingly higher than before the pandemic’. And maybe add that for right-sided voters, despite a significant increase of disagreement during the outbreak, the levels were restored during the outbreak.

MINOR CONCERNS AND SUGGESTIONS

- Table S5 is a bit too crowded. I found a bit difficult to unpack the results from it. I would personally

find more accessible a figure composed of 12 panels (subplots). Each panel could include either five groups of three bars (one group per degree of agreement), or three groups of five bars (one group per wave). Although both options are equally informative, I would personally go for the latter.

- In addition, the evolution across waves of the average score of each proposition (sum of proportion of respondent times the level of agreement) could be shown.

- I have nothing to object to the analysis of the ‘transcendental’ questions (items 2 and 8), as the p values are the most reliable (or neutral) indicator of the existence of an effect. However, I think that the effect highlighted in the abstract is less remarkable than what visually stands out in Fig 1, and is pointed out in lines 331-333 of the discussion: during the outbreak the proportion of respondents that believed in an afterlife and in God answering people’s prayers decreased, while during de-escalation the tendency was reversed with a comparable strength. In the discussion it is reviewed how different groups may react differently to the pandemic, but I would suggest a discussion of the general effect I have just highlighted. To me, this could point to some interesting psychological effects, for example related to pessimism and nihilism during outbreak and lockdown, followed by optimism during de-escalation and drop in the number of deceased (many other interpretations are equally or more acceptable).

- Have the authors considered how question polarity (positive or negative wording of the sentence) influences the level of agreement? See for example in [Holleman, B., Kamoen, N., Krouwel, A., Pol, J. V. D., & Vreese, C. D. (2016). Positive vs. negative: The impact of question polarity in voting advice applications. PloS one, 11(10), e0164184.], where It reads that the ‘choice to word questions positively (e.g., ‘The city council should allow cars into the city centre’) or negatively (‘The city council should ban cars from the city centre’) systematically affects the answers’.

For instance, if people is presented with the proposition ‘There is nothing beyond death’, they may be inclined to disagree to a higher extent than the agreement they would show when presented with the proposition ‘There is something after death’. Maybe the authors could comment on the criteria followed to formulate the propositions with a positive or a negative phrasing.

- Line 75: ‘At the time these lines are written’. For the sake of the reader’s curiosity, the (approximate) date could be stated.

- Line 211-213: ‘strongly agreed’ is used in the beginning of the sentence to describe the preference of those who either agreed or strongly agreed. A more accurate rewording could be ‘overwhelmingly agreed’.

- In lines 342-345 the attitude towards science is said to be partially critical because there is weaker ‘agreement with the capacity of science to achieve immortality’. But this seems more a question that evaluates faith in a very unlikely achievement, on which tiny proportion of researches may be working. I think that a more realistic question would have been about achieving perfect health until death, or extending life for centuries. Having said this, I am personally astonished by the high proportion of respondents that think that science will achieve immortality!

- In line 386, I am not sure that non-native English readers will understand the meaning of the word ‘cold’ in the context of the sentence.

- I have particularly enjoyed the paragraph starting in line 388. It is a clear exposition of how beliefs shape our attitudes and behaviours, and how understanding the ways in which traumatic global-scale events affect beliefs should be a priority of governments in the circumstance of adopting policies that involve public health and fundamental rights.

- In line 396, ‘Trump believers’ are mentioned. Maybe a very brief explanation of what this collective is would be adequate.

- Is the proportion of left and right-sided voters among the respondents similar to that of the total Spanish population? Could this have an influence on the results (for example Fig 1 or table S5)? An interesting test would be to add a newer version of Fig 1 and table S5, but correcting to the actual proportions in Spain, obtained from national surveys or the results of previous general elections.

- Regarding the title of papers, I personally find more powerful to capture in them the most striking result of the paper. In the case of the work discussed here, I could go for something like ‘Polarization on the belief system as a consequence of the COVID-19 pandemic: the case of Spain’.

- The longitudinal data are very interesting, and they well deserve a (supplementary) figure, similar to Figs 1 or 2.

- In table S5 the specific phrasing of the degrees of agreement is presented. Is not the use of ‘irrefutable’ a bit contradictory with the fact of not changing opinion? It remains me of the ‘Irresistible force paradox’ (What happens when an unstoppable force meets an immovable object?). Why the authors did not use a similar wording and phrasing than in degrees 2 and 4. For example ‘I agree, and I would not change my mind even if I were shown strong evidence’.

- Is the list of the 90 questions that comprised the original survey, and the answers to all of them, available somewhere else? If not, will they be available/published in the future? It would be very interesting for the reader of this paper to have access to them.

- In line 504, Statistical analyses, it reads ‘(…) selected dependent variables could significantly predict (…)’. I wonder whether the authors meant ‘independent variables’ instead.

- In line 521, Tables 3 and 4 are referenced regarding the ‘differences in predicted probabilities’. May be the authors actually referring to the ‘contrast’ results in Table S6?

6. PLOS authors have the option to publish the peer review history of their article (what does this mean?). If published, this will include your full peer review and any attached files.

Reviewer #1: No

Reviewer #2: No

---

## [Author Response · Author response to Decision Letter 0]

18 May 2021

We are extremely grateful to the Academic Editor, Editorial Office personnel and both Reviewers for their valuable and extensive comments and criticisms. In our opinion, most issues exposed important weaknesses of our proposal, and we have carefully dealt with them to improve our manuscript.

ACADEMIC EDITOR REVIEW

AE1: An overall concern that I share with the reviewers is that you find a way to focus your findings and find a more concise, coherent presentation of the results

RESPONSE: Thank you very much for this suggestion of improvement. We have stated more clearly the main (and secondary) goals of our research. Concision has been challenging, because new analyses and clarifications were asked by both Reviewers. However, we think that the main outline of our research is better explained in this new version of the manuscript, and results are more coherently presented.

AE2: Reviewer 1 mentions the placement of the Methods section at the end, and I also found it necessary to read the end before I could understand the results and discussion section.

RESPONSE: The Methods section has been placed after the Introduction. Please note that this is not highlighted by the ‘track changes’ tool, in order to improve readability.

AE3: Another concern is with the sample: please expand on the purpose of the initial study, which used snowball sampling and network connections. It seems that similar nonprobability sampling methods were used for the second and third waves. This is likely the reason for the rather small proportion of older persons in the sample. As one reviewer suggests, please compare the age structure of your sample with that of the nation.

RESPONSE: We appreciate this comment, which led us to re-analyze the data after weighting with iterative proportional fitting (raking). Results are mostly replicated by this new approach. Please find more details about this in our response to comment R2.13 by Reviewer 2.

EDITORIAL OFFICE COMMENTS

EO1. Please ensure that your manuscript meets PLOS ONE's style requirements, including those for file naming. The PLOS ONE style templates can be found at

RESPONSE: To the best of our knowledge, we meet all style requirements in the new version of the manuscript.

EO2.Thank you for stating the following in the Acknowledgments Section of your manuscript:

"This research has been supported by the Institute for Culture and Society (non-competitive funding)"

 "The authors received no specific funding for this work"

RESPONSE: Thanks for clarifying this. We have removed that sentence from the Acknowledgements. There is no need to disclose any funding, so “The authors received no specific funding for this work” is correct. We included the sentence about the Institute for Culture and Society in the previous version because our institution covers our salaries, but that is definitely not ‘specific funding’.

EO3. Please include a copy of Table 4 which you refer to in your text on page 21.

RESPONSE: Thank you for spotting this. We have re-organized Tables and Figures, and hopefully all of them are present and correctly named in the main text and supplementary information.

EO4. Please include captions for your Supporting Information files at the end of your manuscript, and update any in-text citations to match accordingly. Please see our Supporting Information guidelines for more information: http://journals.plos.org/plosone/s/supporting-information.

RESPONSE: Thank you. We have included this at the end of the manuscript, after the References. Also, for each element, the title is typed in bold, and the legend is in regular typeset.

REVIEWER 1

The authors conduct a three-wave survey do examine how the COVID19 pandemic and political affiliation affect a range of beliefs. They have a good grasp of psychological literature and an adequate knowledge of statistical analyses. However, the article suffers from disorganization and lack of focus. I provide detailed comments for each section below.

INTRO

Good intro and historical setup.

R1.1 PREVIOUS RESEARCH

The authors should be clear that this is about “period effects” as distinct from other types of attitude change. H1N1 example is good, perhaps more detail on this as a prior example. The use of shattered assumptions theory is good, and the authors could also mention work in sociology on cultural trauma (Alexander) and natural disasters (Erikson). On the topic of religious coping, Kenneth Pargament’s work would be relevant here. The psych of conspiracy theories doesn’t seem relevant in this section, especially as it doesn’t come up much in remainder of paper. In line 116, “whether a set of beliefs changed,” is a weak research question. In line 120, “will encompass” is a vague hypothesis. Be more specific about hypotheses and expectations.

RESPONSE: We clarify that our research shows “period effects” on beliefs (p. 4; please note that page numbers refers to the document with all track changes active). We appreciate the suggestions about Alexander’s and Erikson’s works, which are now mentioned (p. 5). Pargament’s theory of religious coping is also introduced (p. 6). The reference to conspiracy theories has been removed. Finally, hypotheses and expectations have been more adequately formulated (p. 6): “Our research question is whether the attitude of Spanish residents towards social, spiritual and interpersonal affairs has been affected by the worst global crisis in the last decades. Our hypothesis, following previous research on H1N1 pandemic and social trauma, is that confidence in authorities, transcendental and prosocial beliefs will be strengthened. However, given the actual polarization of Spanish society in terms of political ideology [19], we predict that belief changes will be strongly influenced by individual political preference.”

R1.2 Methods section should come before results. I had to find / read this section before making sense of results.

RESPONSE: The Methods section has been placed after the Introduction. Please note that this is not highlighted by the ‘track changes’ tool, in order to improve readability.

R1.3 SAMPLE

The authors make use of a pre-existing survey to approximate a longitudinal survey in three waves. However, the unbalanced sample sizes should be justified / addressed. Wave 1 is almost 1/10th the size of Wave 2 (156 vs. 1182). The addition of 97 linked respondents between W2 and W3 helps the argument.

RESPONSE: We explain in more detail why the sample sizes of the three waves are so disparate: Materials and Methods, “Samples” subsection (p. 7): “This project started in 2018…”. The different sample sizes are further justified on p. 8. Also, we have stressed the results on the 97 linked respondents, as suggested by both Reviewers (see below).

R1.4 ANALYSIS

The authors use ordinal logistic regression, to test effect of the pandemic, politics, and their interaction on personal beliefs. In line 504, "dependent" should be independent. In line 520-521, why present “differences in predicted probabilities”? These are not present in table 3 as mentioned, and table 4 is missing. In line 521 to 527, was sex not included in main analyses? This section was confusing.

RESPONSE: Thank you for spotting the typo in line 504 of the previous version. It now reads “independent variables” (p. 13 of the revised version). We explain in more detail what “differences in predicted probabilities” mean, which is used to understand in more detail the results of interaction in ordinal regressions (pp. 13-14): “Let us consider, for instance, the interaction between wave and political preference: based on the ordinal logistic regression models, “margins” computes the probability predicted by the model of a participant with certain political preference to show certain degree of disagreement (1 to 5) in certain wave with respect to other time point. For example, regarding item 4 (“I think that government authorities tend to be intrusive and controlling”), the model predicts that a left-sided voter has a significantly higher probability (nearly a 42%) of agreeing with this proposition before the pandemic with respect to the outbreak”. We are very sorry about the confusion on Tables: the reference to some Tables was messed when some of them were moved to the Supplementary Information. All Tables and their mention in the text are (hopefully) correct in this new version. Sex was also included in the main analyses. This subsection has been thoroughly revised in order to improve readability (pp. 13-15).

RESULTS

R1.5 The authors find a handful of significant relationships, but need to do a better job tying them together into a coherent story. This may mean ignoring some significant relationships to highlight substantively related findings. Tables S1-S4 are confusing. Why not present all covariates at once? No need to present four tables for one regression analysis. Does figure 1 come from Table S5? If so, the figure should depict significant differences somehow. Line 185-188 is unclear: chi-squared tests on what propositions? “analyzed residuals”? This procedure needs clarification.

RESPONSE: Once again, we are grateful to the Reviewer for pointing to unclear analyses or results. In the Supplementary Information, we now include a large table (S2 Table) that shows the results of the ordinal regressions. In the main text, we now stress that our main independent variables of interest are wave and politics, but in this supplementary table we report the contribution of all variables to the model. In this new version, line graphics have been substituted for stacked bars, which give a more precise information and reduce noise. In our opinion, it is still important to provide (old) Table S5 (now S3 Table), even though data overlap with Figure 1. We explain in more detail the chi-squared tests that were carried out (S1 Text, pp. 6-8): “In the main text, we present the change in the endorsement of the 12 propositions as a consequence of the COVID-19 pandemic. In order to further explore these differences, for every item, we statistically compared the percentage of participants that responded each disagreement level (1 to 5) at each wave (see Table S3). Thus, chi-squared tests were performed for each item of the survey using the ‘tabulate’ command in Stata. The null hypothesis of this test is that the proportion of responses to each disagreement level is unchanged throughout the three waves. Since this chi-squared test provides a single significance value (for each item of the survey), we analyzed the adjusted residuals with the ‘tabchi’ command in order to detect the main contributors to the significant results. This command provides observed and expected frequencies, as well as adjusted residuals. Data are summarized in Table S3, and the results of both commands (‘tabulate’ and ‘tabchi’) are uploaded as Supplementary Information. We present here the main contributors to each significant overall result. With respect to item 1…”.

R1.6 Figure 2 is interesting, but there are many redundant lines. Is there a way to depict overall opinion rather than showing mirrored agree/disagree lines? ie. If fewer people agree with a statement, more will disagree. As-is this graph is hard to interpret. Table 2 is good.

RESPONSE: We completely agree about Figure 2. As mentioned, we present stacked bars graphics in the new version of the manuscript.

R1. 7 DISCUSSION

The authors find that during the outbreak, more people accepted government control and skewed towards group priorities over individual priorities. During de-escalation, these feelings waned.

I recommend adding justifications / clarifications as described above, and streamlining the discussion of results to focus on main findings. Best of luck to the authors.

We truly appreciate the Reviewer’s suggestions/concerns.

REVIEWER 2

This work presents the result from a sequential survey conducted in Spain across three temporal waves: one year before the COVID-19 pandemic, at its outbreak, and during de-escalation. Respondents were asked to declare their level of agreement with 12 propositions concerning transcendental, social or interpersonal subjects. The aim of the authors was to investigate the effect of the pandemic on the beliefs related to the propositions. They also investigated whether the effect was influenced by some features of the respondents: age, gender, civil status, political ideology, having an acquaintance affected of COVID-19, and a COVID-19 deceased relative.

The authors claim that:

- Despite the lockdown, respondents tolerated being controlled by authorities, and acknowledged the importance of group necessities over individual rights.

- De-escalation changed the above beliefs.

- Transcendental beliefs were strengthened.

- Left-sided voters did not see authorities as intrusive

- Transcendental beliefs prevailed among right-sided voters.

- The pandemic resulted in a polarization of belief system based on political ideology.

GENERAL EVALUATION

The study presented here is of the highest relevance. On the one hand, psychologists and sociologists will gain knowledge about the impact of events like the COVID-19 pandemic on the belief system of individuals. On the other hand, politicians and international organizations may use results of this kind to modulate their policies according to their expected impact on the population. I would really like to see studies like this one performed in other countries and when facing other traumatic events.

The paper is written in a clear way, with an accurate yet easy to follow English. The authors have made a big effort to dissect the results of the survey and extract from them relevant and potentially beneficial information regarding the effect of the pandemic on beliefs. I think that the claims made in the abstract are well addressed, and they are well supported by the results of the survey. 

However, I am going to express two concerns about the main claims made by the authors. I will discuss other minor concerns and suggestions that, if addressed, I think will enrich this already superb work. 

MAIN CONCERNS

R2.1 In the abstract, the authors claim that ‘left-sided voters did not see authorities as intrusive’. But in light of Fig 2, item 4, and Table S6, I think it is more accurate to say that the proportion of left-sided voters that agreed with the proposition ‘Government authorities are intrusive’ decreased dramatically, and the proportion that disagreed had the opposite trend (which is a striking result by itself). But still the level of agreement was slightly higher (or at least not any lower) than the level of disagreement.

RESPONSE: Thank you very much for noticing this. The Reviewer is absolutely right: most of left-sided voters did see authorities as intrusive in the outbreak and de-escalation, although the proportion of them who did so greatly decreased. We have reworded the Abstract and the Discussion (p. 31; please note that page numbers refer to the manuscript with ‘all track changes’ active).

R2.2 Similarly, in lines 360 and 361 of the Discussion, the authors claim that ‘during de-escalation, left-sided voters found tolerable being controlled by others’. But in lines 225-228 of the Results, as well as in Fig 2, item 10, what we see is that the proportion of left voters that agreed with the proposition ‘Being controlled by others is intolerable’ decreased dramatically (which, again, is a remarkable result). But still the level of agreement with the intolerability of being controlled by others is much higher. 

In the two above cases, I think that slight rephrasing of the sentences might be enough. For example:

• ‘a remarkably higher proportion of left-sided voters did not see authorities as intrusive after the outbreak’.

• ‘during de-escalation, the proportion of left-sided voters that found tolerable being controlled by others remained overwhelmingly higher than before the pandemic’. And maybe add that for right-sided voters, despite a significant increase of disagreement during the outbreak, the levels were restored during the outbreak.

RESPONSE: Indeed, this is also the case. We have reworded the Discussion (p. 32).

MINOR CONCERNS AND SUGGESTIONS

R2.3 Table S5 is a bit too crowded. I found a bit difficult to unpack the results from it. I would personally find more accessible a figure composed of 12 panels (subplots). Each panel could include either five groups of three bars (one group per degree of agreement), or three groups of five bars (one group per wave). Although both options are equally informative, I would personally go for the latter. 

RESPONSE: Inspired by this suggestion, we have substituted line graphs for stacked bars which, in our opinion, are more clear. We agree that Table S5 (now S3 Table) is too crowded, but in our opinion it is important to report the proportions of all disagreement levels by wave. We understand it would be a terrible table for the main text, but we hope it is acceptable as supplementary information.

R2.4 In addition, the evolution across waves of the average score of each proposition (sum of proportion of respondent times the level of agreement) could be shown. 

RESPONSE: We did not think about this particular display of our results, but it looks very informative. We have included this figure as Supplementary Information (S1 Figure).

R2.5 I have nothing to object to the analysis of the ‘transcendental’ questions (items 2 and 8), as the p values are the most reliable (or neutral) indicator of the existence of an effect. However, I think that the effect highlighted in the abstract is less remarkable than what visually stands out in Fig 1, and is pointed out in lines 331-333 of the discussion: during the outbreak the proportion of respondents that believed in an afterlife and in God answering people’s prayers decreased, while during de-escalation the tendency was reversed with a comparable strength. In the discussion it is reviewed how different groups may react differently to the pandemic, but I would suggest a discussion of the general effect I have just highlighted. To me, this could point to some interesting psychological effects, for example related to pessimism and nihilism during outbreak and lockdown, followed by optimism during de-escalation and drop in the number of deceased (many other interpretations are equally or more acceptable). 

RESPONSE: We have slightly reworded the Abstract to be more precise about this effect (“Besides, transcendental beliefs –God answering prayers and the existence of an afterlife– declined after the outbreak, but were strengthened in the de-escalation”). 

R2.6 Have the authors considered how question polarity (positive or negative wording of the sentence) influences the level of agreement? See for example in [Holleman, B., Kamoen, N., Krouwel, A., Pol, J. V. D., & Vreese, C. D. (2016). Positive vs. negative: The impact of question polarity in voting advice applications. PloS one, 11(10), e0164184.], where It reads that the ‘choice to word questions positively (e.g., ‘The city council should allow cars into the city centre’) or negatively (‘The city council should ban cars from the city centre’) systematically affects the answers’.

For instance, if people is presented with the proposition ‘There is nothing beyond death’, they may be inclined to disagree to a higher extent than the agreement they would show when presented with the proposition ‘There is something after death’. Maybe the authors could comment on the criteria followed to formulate the propositions with a positive or a negative phrasing. 

RESPONSE: Thank you for this suggestion, which we did not consider. First of all, since each item is analyzed independently, the effect of question polarity in the formulation of the item should not be an important factor. Moreover, the main goal is to compare (dis)agreement at three different time points, and every item is formulated in the same way throughout the three waves. In any case, the Reviewer is probably right on a possible increased disagreement towards propositions formulated with negative wording (item 2). However, this would not affect the main objective of our research, which is a different endorsement of this proposition before and during the pandemic (outbreak and de-escalation). In any case, we have looked for similar surveys performed by the Spanish Center for Sociological Research (CIS), and we found that between October 2017 and January 2018 (study #3194; N=1733) the following question (among others) was asked: “Do you believe in life after death?” 17.9% answered “absolutely yes”, 23.7 answered “probably yes”, 18.7 answered “probably not”, and 29.6 answered “absolutely not”. This would point to the accuracy of the Reviewer’s comment, since according to our data 41.2% of participants strongly disagree with the non-existence of life after death, and only 10.6% strongly agree with this non-existence. We have included this putative limitation in the new version of the Discussion (p. 33).

R2.7 Line 75: ‘At the time these lines are written’. For the sake of the reader’s curiosity, the (approximate) date could be stated. 

RESPONSE: We expected to update these data in the revised version of the manuscript (by the way, the information shown in the previous version was from October 2020, if we remember correctly). However, we realized that it would be more informative (for the sake of our research) to explain the situation when data were collected (pp. 3-4), and not when the paper was written. Spain is still one of the most affected countries in the world, but what happened after collecting our data should be irrelevant for our research. We hope the Reviewer agrees with this new approach. 

R2.8 Line 211-213: ‘strongly agreed’ is used in the beginning of the sentence to describe the preference of those who either agreed or strongly agreed. A more accurate rewording could be ‘overwhelmingly agreed’. 

RESPONSE: Thank you. We have corrected this (p. 23).

R2.9 In lines 342-345 the attitude towards science is said to be partially critical because there is weaker ‘agreement with the capacity of science to achieve immortality’. But this seems more a question that evaluates faith in a very unlikely achievement, on which tiny proportion of researches may be working. I think that a more realistic question would have been about achieving perfect health until death, or extending life for centuries. Having said this, I am personally astonished by the high proportion of respondents that think that science will achieve immortality! 

RESPONSE: The Reviewer is definitely right. We have reworded this part of the Discussion to clarify this ‘realistic support’ (p. 31).

R2.10 In line 386, I am not sure that non-native English readers will understand the meaning of the word ‘cold’ in the context of the sentence. 

RESPONSE: Now we use the expression “conscious endorsement of a set of propositions” (p. 33).

R2.11 I have particularly enjoyed the paragraph starting in line 388. It is a clear exposition of how beliefs shape our attitudes and behaviours, and how understanding the ways in which traumatic global-scale events affect beliefs should be a priority of governments in the circumstance of adopting policies that involve public health and fundamental rights. 

RESPONSE: Thank you for this positive and encouraging comment.

R2.12 In line 396, ‘Trump believers’ are mentioned. Maybe a very brief explanation of what this collective is would be adequate. 

RESPONSE: In the revised version of the manuscript, we use the expression “US citizens holding faith in Trump”, as used in the original research. We also provide an example to clarify what the authors of that research mean (pp. 34).

R2.13 Is the proportion of left and right-sided voters among the respondents similar to that of the total Spanish population? Could this have an influence on the results (for example Fig 1 or table S5)? An interesting test would be to add a newer version of Fig 1 and table S5, but correcting to the actual proportions in Spain, obtained from national surveys or the results of previous general elections. 

RESPONSE: This comment led us to re-analyze the data, correcting our database for the actual proportion of the Spanish population in terms of age, sex and political preference. This is explained in detail in Materials and Methods (Table 1, pp. 9-10). First, we have included in Table 1 a column with the proportion of Spanish residents in terms of sex, age group and voting preference according to the latest national elections (November 10, 2019). Demographic national data are extracted from the census, updated on July 2020. We explain the following on pp. 9-10: “Statistical analyses were carried out on ‘raw’ data as described below. Besides, in order to correct the unbalance in sex, age and political preference of our sample with respect to the nation totals, analyses were also replicated in a weighted database.”. Then, in the Supplementary Methods of S1 Text (pp. 2-3), we explain the following: “In detail, each single respondent was assigned a weight to correct over or underrepresentation of sociodemographic variables. We used iterative proportional fitting (i.e. raking) for this purpose, by means of ‘ipfraking’ tool in Stata [1]. There were two control variables: 1) a combination of sex and age group, 2) political preference”. We provide the data that was taken as reference for weighting our database by controlling for sex/age, and political preference. We end up this section with the following text: “In conclusion, this procedure assigns a weight to each participant in order to correct their under or overrepresentation in the sample. Analyses on the weighted database are performed in Stata, in general terms, with prefix ‘svy:’, after establishing the survey parameters with ‘svyset’”.

We have included a new section in Results, where the main results are mostly replicated with weighted data. We have also included new supplementary figures and tables to show the similarities (and differences) between analyses on raw and weighted data.

R2.14 Regarding the title of papers, I personally find more powerful to capture in them the most striking result of the paper. In the case of the work discussed here, I could go for something like ‘Polarization on the belief system as a consequence of the COVID-19 pandemic: the case of Spain’. 

RESPONSE: Thank you for the suggestion. We have changed the title of the manuscript.

R2.15 The longitudinal data are very interesting, and they well deserve a (supplementary) figure, similar to Figs 1 or 2. 

RESPONSE: We appreciate this valuable advice, which was also remarked by Reviewer 1. We have included new figures on the longitudinal results, remarkably histograms depicting individual changes in agreement (S3 Figure).

R2.16 In table S5 the specific phrasing of the degrees of agreement is presented. Is not the use of ‘irrefutable’ a bit contradictory with the fact of not changing opinion? It remains me of the ‘Irresistible force paradox’ (What happens when an unstoppable force meets an immovable object?). Why the authors did not use a similar wording and phrasing than in degrees 2 and 4. For example ‘I agree, and I would not change my mind even if I were shown strong evidence’. 

RESPONSE: We understand this is a controversial topic that we try to avoid in this manuscript. In a recent theoretical publication (Camina et al. 2020. Foundations of Science), we operationalize beliefs in a rigorous and restricted way. As we have included in the revised version of the current manuscript (pp. 11-12), “According to our theoretical framework [21], a belief is: (1) a proposition that is taken to be true; and (2) which the subject is willing to hold even if irrefutable evidence were hypothetically argued against it. In the current study, believing in a proposition is equivalent to expressing a strong agreement with it (answering 1), and a belief in the negation of the proposition is the same as a strong disagreement with it (answering 5). Finally, having an opinion for or against a proposition is equivalent to expressing agreement (answering 2) or disagreement (answering 4) with it”. As we said above, we intend to skip the debate on this operationalization in the current manuscript by talking about “degree of (dis)agreement”. The fact that many of the volunteers of the current project (longitudinal dataset) changed their beliefs (in a strict sense: they declared in the outbreak that they would not change their mind even in light of hypothetical irrefutable proof against (1) or for (5) the proposition; however, they did change their mind) during de-escalation demonstrates that our view is plausible and beliefs are not immutable, even though subjects may think they are so when they hold them.

R2.17 Is the list of the 90 questions that comprised the original survey, and the answers to all of them, available somewhere else? If not, will they be available/published in the future? It would be very interesting for the reader of this paper to have access to them. 

RESPONSE: We are about to submit the manuscript where we show the representation of the belief system with graph theory. That manuscript will include the 90-item survey and individual responses. We believe that including that information here may be confusing for the reader.

R2. 18 In line 504, Statistical analyses, it reads ‘(…) selected dependent variables could significantly predict (…)’. I wonder whether the authors meant ‘independent variables’ instead. 

RESPONSE: Thank you for spotting this typo, which has been corrected.

R2. 19 In line 521, Tables 3 and 4 are referenced regarding the ‘differences in predicted probabilities’. May be the authors actually referring to the ‘contrast’ results in Table S6? 

RESPONSE: Once again, we are grateful to both Reviewers for detecting these errors in the manuscript. Some tables were erroneously cited in the text when moving some of them between the main text and the Supplementary Information. We have corrected this in the new version of the manuscript. With respect to predicted probabilities and ‘contrast’ results, as per Reviewer 1’s request, we have explained more clearly these analyses (pp. 13-14): “Let us consider, for instance, the interaction between wave and political preference: based on the ordinal logistic regression models, “margins” computes the probability predicted by the model of a participant with certain political preference to show certain degree of disagreement (1 to 5) in certain wave with respect to other time point. For example, regarding item 4 (“I think that government authorities tend to be intrusive and controlling”), the model predicts that a left-sided voter has a significantly higher probability (nearly a 42%) of agreeing with this proposition before the pandemic with respect to the outbreak”.

---

## [Decision Letter · Decision Letter 1]

8 Jun 2021

PONE-D-20-35324R1

Polarization of beliefs as a consequence of the COVID-19 pandemic: the case of Spain

PLOS ONE

Dear Dr. Bernácer,

Thank you for submitting your manuscript to PLOS ONE. After careful consideration, we feel that it has merit but does not fully meet PLOS ONE’s publication criteria as it currently stands. Therefore, we invite you to submit a revised version of the manuscript that addresses the points raised during the review process.

The reviewers have completed their reviews of your revised manuscript and they agree that it is substantially improved.  Thank you for your extremely careful attention to their many comments and suggestions.  Reviewer 2 notes just a small typo. 

Reviewer 1, however is questioning the analysis represented and interpreted in Table 2.  The footnote and text say that the analyses include an interaction term for wave x political preference.  If that is the case, then the main effects of wave should not be interpreted.  Reviewer 2 suggests running the same analyses and presenting them both with and without the interaction, and I agree.  Reviewer 2 has some additional minor suggestions -- please attend to these edits.

In addition to the reviewers' comments, I would add that lines 302-309 on p. 13 constitute results, and therefore belong in the results section. Also, there is a typo in the note to Table 3, line 397 - "that" should be "than".

You may consider this a "conditional accept".  I will accept the manuscript upon receipt of the revisions/responses.

We look forward to receiving your revised manuscript.

Kind regards,

Ellen L. Idler

Academic Editor

PLOS ONE

Journal Requirements:

Additional Editor Comments (if provided):

Reviewers' comments:

Reviewer's Responses to Questions

**Comments to the Author**

1. If the authors have adequately addressed your comments raised in a previous round of review and you feel that this manuscript is now acceptable for publication, you may indicate that here to bypass the “Comments to the Author” section, enter your conflict of interest statement in the “Confidential to Editor” section, and submit your "Accept" recommendation.

Reviewer #1: (No Response)

Reviewer #2: All comments have been addressed

2. Is the manuscript technically sound, and do the data support the conclusions?

Reviewer #1: Yes

Reviewer #2: Yes

3. Has the statistical analysis been performed appropriately and rigorously? 

Reviewer #1: No

Reviewer #2: Yes

4. Have the authors made all data underlying the findings in their manuscript fully available?

Reviewer #1: Yes

Reviewer #2: Yes

5. Is the manuscript presented in an intelligible fashion and written in standard English?

Reviewer #1: Yes

Reviewer #2: Yes

6. Review Comments to the Author

Reviewer #1: Overall

- This revision is a significant improvement on the original submission. I appreciate the inclusion of a research question and hypothesis in the introduction. The research methods are also described clearly and I arrived at the results section with a much better idea of the data and plan for analysis. The figures were also much easier to interpret. However, there remains one major issue and a few points of clarification before this manuscript can be accepted. Note: all page numbers refer to document without track changes.

Major issue

- The first models presented in Fig 1, Table 2, and S2 Table talk about the effect of wave on beliefs. However, based on the sentence that starts on p14 line 342 and the caption to Table 2, it sounds like these models also include the interaction of political preference and wave. If correct, the inclusion of an interaction term radically changes the interpretation of any “main effect” of wave on belief. As currently specified, I think the main effects for wave only hold when politics = 0 (“right voter”) and thus do not convey the general effect of the pandemic across all voters. The easiest solution would be to simply rerun the models without the interaction effect (reproducing Fig 1, Table 2, and S2 Table) and then include the interaction effect for the subsequent discussion of results that begins on p17 “Effect of political preferences on pandemic-related belief changes.”

Minor notes

- p4 line 93: extra period.

- p10 line 243: avoid using “degree” when describing agreement or disagreement. “Level” would work better here. Same issue on p14 line 339.

- p11: operationalization of belief vs opinion seems to depart from conventional uses of the word “belief” but will trust the cited source here.

- I still need help reading Table 3 and Table 4. Are the “contrast” values the changes in predicted probability? I had trouble moving between Table 4 and the main text on p25 line 561-562: where does the 40% to 78.3% come from? An extra sentence or two of clarification are needed here.

- p26 line 601: if this is the “first report” on belief change and COVID-19, what about citation #24?

- p29 line 676-678: study should be properly cited in reference section.

Reviewer #2: The authors have done an awesome work on addressing the concerns expressed by the reviewers. I think that the paper is ready to be published as it stands now. It will be an extremely valuable contribution and will help understand the psychological and sociological foundations of beliefs.

P.S. I think there is a typo in line 671 of the Discussion (version without track changes active). It reads ‘other’ when it should read ‘others’.

7. PLOS authors have the option to publish the peer review history of their article (what does this mean?). If published, this will include your full peer review and any attached files.

Reviewer #1: No

Reviewer #2: No

---

## [Author Response · Author response to Decision Letter 1]

24 Jun 2021

Once again, we are extremely grateful to the Academic Editor and Reviewers for their helpful comments, which we fully address in the revised version of our manuscript as follows.

ACADEMIC EDITOR:

Reviewer 1, however is questioning the analysis represented and interpreted in Table 2. The footnote and text say that the analyses include an interaction term for wave x political preference. If that is the case, then the main effects of wave should not be interpreted. Reviewer 2 suggests running the same analyses and presenting them both with and without the interaction, and I agree. Reviewer 2 has some additional minor suggestions -- please attend to these edits.

Response: We have included all suggestions raised by Reviewer 2 (see below). We are extremely grateful for having detected this misinterpretation of our analyses. Please note that results do not substantially change (Table 2; changes in pp. 12-13 (Methods), and pp. 14, 15 and 17 in Results; page numbers refer to text with track changes). We have also corrected the analyses pertaining participants with a COVID-19 sick acquaintance, since they had the same problem (pp. 20-22).

In addition to the reviewers' comments, I would add that lines 302-309 on p. 13 constitute results, and therefore belong in the results section.

Response: We have removed these lines from the Methods (p. 13).

Also, there is a typo in the note to Table 3, line 397 - "that" should be "than".

Response: Thanks. This typo has been corrected.

In addition, we have referenced two supplementary materials that were missing in the previous version of the manuscript (that is, they were correctly uploaded, but they were not in the list of Supplementary Information at the end of the manuscript): S1 Datasets and S1 Statistical reports (p. 38). We have also corrected some typos.

Reviewer #1: Overall

- This revision is a significant improvement on the original submission. I appreciate the inclusion of a research question and hypothesis in the introduction. The research methods are also described clearly and I arrived at the results section with a much better idea of the data and plan for analysis. The figures were also much easier to interpret. However, there remains one major issue and a few points of clarification before this manuscript can be accepted. Note: all page numbers refer to document without track changes.

Major issue

- The first models presented in Fig 1, Table 2, and S2 Table talk about the effect of wave on beliefs. However, based on the sentence that starts on p14 line 342 and the caption to Table 2, it sounds like these models also include the interaction of political preference and wave. If correct, the inclusion of an interaction term radically changes the interpretation of any “main effect” of wave on belief. As currently specified, I think the main effects for wave only hold when politics = 0 (“right voter”) and thus do not convey the general effect of the pandemic across all voters. The easiest solution would be to simply rerun the models without the interaction effect (reproducing Fig 1, Table 2, and S2 Table) and then include the interaction effect for the subsequent discussion of results that begins on p17 “Effect of political preferences on pandemic-related belief changes.”

Response: We are extremely grateful for noticing this. Indeed, “main effects” are not interpretable when there is an interaction term involving one or several of the variables included in the interaction. As the Reviewer suggests, the “main effects” we were reporting only referred to politics=0, that is, to right-sided voters. We followed the solution suggested by the Reviewer, and repeated the analyses without the interaction term for that section of the Results. Thus, numbers in Table 2 have been completely changed (we have not used track changes for that). Please note that results have not changed substantially. In fact, just a few changes were necessary in the text (changes in pp. 12-13 (Methods), and pp. 14, 15 and 17 in Results; page numbers refer to text with track changes). We have also corrected the analyses pertaining participants with a COVID-19 sick acquaintance, since they had the same problem (pp. 20-22).

Minor notes

- p4 line 93: extra period.

Response: This has been corrected

- p10 line 243: avoid using “degree” when describing agreement or disagreement. “Level” would work better here. Same issue on p14 line 339.

Response: Thanks for this suggestion. We have substituted all instances of “degree” (in this context) for “level”.

- p11: operationalization of belief vs opinion seems to depart from conventional uses of the word “belief” but will trust the cited source here.

Response: Thank you.

- I still need help reading Table 3 and Table 4. Are the “contrast” values the changes in predicted probability? I had trouble moving between Table 4 and the main text on p25 line 561-562: where does the 40% to 78.3% come from? An extra sentence or two of clarification are needed here.

Response: We understand the confusion. As expressed in the footnotes of Tables 3 and 4, “contrast” refers to the change in the predicted probability to respond 1 (columns on the left) or 5 (columns on the right). We decided to display only these results for the sake of clarity. We have highlighted this part of the footnote by adding an asterisk on “Contrast” in both tables. However, the percentages mentioned by the Reviewer do not come from the changes in the predicted probabilities: those are ‘actual’ (and not predicted) percentages of the longitudinal data, which are not directly shown in any table or figure (although they can be easily calculated from the datasets). The Reviewer’s confusion is absolutely justified, and this comment is a good opportunity to clarify our results. First of all, we now include the Stata log with these description (longitudinal dataset: proportion of each disagreement level by wave by politics, for each item). This is included in S1 Statistical Reports. Second, we clearly explain where those percentages are coming from: “Also, the percentage of respondents that expressed each disagreement level for every item is shown in S1 Statistical reports, sorted by political ideology (right-sided voter, yes/no) and wave (outbreak/de-escalation). With regards to these data,…” (p. 26). Finally, changes in predicted probabilities are more clearly explained: “This effect was also found in the predicted probabilities of the multilevel model shown in Table 4: for right-sided voters, there was a 19.3% increased probability of strongly agreeing with authorities being intrusive in the de-escalation with respect to the outbreak; however, for non-right-sided voters, there was a 12.3% decreased probability of strongly agreeing in the de-escalation than in the outbreak” (pp. 26-27). 

- p26 line 601: if this is the “first report” on belief change and COVID-19, what about citation #24?

Response: This is correct. This sentence has been rewritten: “The main novelty of this research is to assess belief changes due to the COVID-19 pandemic across several domains, which is especially valuable for having been carried out in one of the first and most affected countries.” (p. 28). 

- p29 line 676-678: study should be properly cited in reference section.

Done (p. 31).

Reviewer #2: The authors have done an awesome work on addressing the concerns expressed by the reviewers. I think that the paper is ready to be published as it stands now. It will be an extremely valuable contribution and will help understand the psychological and sociological foundations of beliefs.

P.S. I think there is a typo in line 671 of the Discussion (version without track changes active). It reads ‘other’ when it should read ‘others’.

Response: Thank you for noticing this. We have corrected this typo.

---

## [Editor Report · Decision Letter 2]

29 Jun 2021

Polarization of beliefs as a consequence of the COVID-19 pandemic: the case of Spain

PONE-D-20-35324R2

Dear Dr. Bernácer,

We’re pleased to inform you that your manuscript has been judged scientifically suitable for publication and will be formally accepted for publication once it meets all outstanding technical requirements.

Kind regards,

Ellen L. Idler

Academic Editor

PLOS ONE

Additional Editor Comments (optional):

Thank you for your careful attention to the expert comments of the reviewers.  The paper is much improved and will be a real contribution to understanding the effects of the pandemic on political and religious beliefs.
---

## [Editor Report · Acceptance letter]

1 Jul 2021

PONE-D-20-35324R2 

Polarization of beliefs as a consequence of the COVID-19 pandemic: the case of Spain 

Dear Dr. Bernacer:

I'm pleased to inform you that your manuscript has been deemed suitable for publication in PLOS ONE. Congratulations! Your manuscript is now with our production department. 

Kind regards, 

on behalf of

Professor Ellen L. Idler 

Academic Editor

PLOS ONE